# Hankel Singular Value Regularization for Highly Compressible State Space Models

**Paul Schwerdtner**
Courant Institute of Mathematical Sciences
New York University
New York, NY 10012
`paul.schwerdtner@nyu.edu`

**Jules Berman**
Courant Institute of Mathematical Sciences
New York University
New York, NY 10012
`jmb1174@nyu.edu`

**Benjamin Peherstorfer**
Courant Institute of Mathematical Sciences
New York University
New York, NY 10012
`pehersto@cims.nyu.edu`

## Abstract

Deep neural networks using state space models as layers are well suited for long-range sequence tasks but can be challenging to compress after training. We use that regularizing the sum of Hankel singular values of state space models leads to a fast decay of these singular values and thus to compressible models. To make the proposed Hankel singular value regularization scalable, we develop an algorithm to efficiently compute the Hankel singular values during training iterations by exploiting the specific block-diagonal structure of the system matrices that we use in our state space model parametrization. Experiments on Long Range Arena benchmarks demonstrate that the regularized state space layers are up to $10\times$ more compressible than standard state space layers while maintaining high accuracy.

## 1 Introduction

### 1.1 Compressing state space models

As deep neural networks (DNNs) get bigger, compression via quantization [27, 32], pruning [64, 39, 62], and distillation [29, 43] becomes ever more important [9, 65, 19]. In this work, we focus on compressing neural networks for long-sequence modeling. In particular, we focus on DNNs with state space models (SSMs) as layers [21, 22, 24, 15], which have been shown to achieve state-of-the-art accuracy while reducing training and inference costs compared to transformer models [21, 53, 59]. Significant progress was necessary to make SSM layers competitive for long-sequence tasks [22, 15]. After the initial success of S4 layers [22], a big step forward were S5 layers [53], which contain linear time-invariant systems that have diagonal system matrices and reach state of the art performance while balancing expressivity with training and inference costs. Additionally, initialization of SSMs has been identified as being critical for training success, for which HiPPO matrices are now commonly used [22].

In this work, we build on DNNs with SSM layers that are well suited for long-sequence tasks and aim to further reduce inference costs by training neural networks such that the SSM layers are highly compressible: The critical quantity in SSMs that controls inference costs is the order $n$, which is the dimension of the internal state [26]. Therefore, instead of training DNNs with general SSMs

as layers with order $n$, we regularize the training to determine SSMs of order $n$ but such that at the same time SSMs with lower order $r \ll n$ exist that mimic the sequence-to-sequence map of the individual SSM layers. Thus, in our approach, training is performed with a larger order $n$ to provide the opportunity for exploring a large parameter space for high expressivity while enabling compression as a post-processing step. Our proposed regularization is founded on well-established and rigorous system-theoretic results [1].

## 1.2 System-theoretic perspective on SSM compressibility

A classical system-theoretic way [1] of describing a sequence-to-sequence map $\{u_k\}_{k=0}^{\infty} \mapsto \{y_k\}_{k=0}^{\infty}$ induced by a linear time-invariant dynamical system (as used in S5 [53]) is via the convolution with the impulse response $h_k \in \mathbb{R}^{p \times m}$,

$$y_k = \sum_{i=0}^{k} h_{k-i} u_i, \qquad k = 0, 1, 2, 3, \dots,$$

where the input $u_k \in \mathbb{R}^m$ and output $y_k \in \mathbb{R}^p$ at time step $k$ are of dimension $m$ and $p$, respectively. Note that the common design choice in deep state space models is to set $m = p$. This convolution leads to the Hankel operator $\mathcal{H} : \ell_2^m \to \ell_2^p$ that acts on sequences and maps $\{u_k\}_{k=0}^{\infty}$ to $\{y_k\}_{k=0}^{\infty}$. The Hankel operator can be explicitly described by the blocks $\mathcal{H}_{ij} = h_{k-i-j}$ for $i, j \in 0, 1, 2, 3, \dots$ and is linear and bounded. Importantly, the number of non-zero singular values of a Hankel operator is finite and gives the McMillan degree, which is the minimal order $n$ necessary for an SSM to describe the map $\{u_k\}_{k=0}^{\infty} \mapsto \{y_k\}_{k=0}^{\infty}$ [1]. Analogous definitions hold for finite-length sequences $n_s < \infty$, in which case the finite Hankel operator is obtained by $H_{ij} = h_{k-i-j}$ for $i, j = 0, \dots, n_s$. While clearly a system of order $n_s$ exists to describe the map $\{u_k\}_{k=0}^{n_s} \mapsto \{y_k\}_{k=0}^{n_s}$, the goal is finding a system with order $n \ll n_s$ that achieves the same mapping or at least a good approximation of it.

The key for compressibility are the singular values of the Hankel operator: if $n$ is the number of non-zero singular values $\sigma_1, \dots, \sigma_n > 0$, then there exists a system of order $r \le n$ that maps $\{u_k\}_{k=0}^{n_s} \mapsto \{\hat{y}_k\}_{k=0}^{n_s}$ with error $\|\hat{y}_k - y_k\|_{\ell_2} \le 2\|u\|_{\ell_2} \sum_{i=r+1}^{n} \sigma_i$. Thus, the quicker the singular values of the Hankel operator decay, the more compressible a system is, which is relevant for, e.g., model reduction [50, 5, 36].

When training SSM layers with standard optimization, however, one only prescribes an order $n$, and then the optimizer can distribute the Hankel singular values at will, which typically leads to systems that are only poorly compressible. In Figure 1, we show the decay of the Hankel singular values of the systems learned for the Long Range Arena (LRA) [59] image benchmark. The Hankel singular values decay slowly and thus there cannot exist a much smaller system with $r \ll n$ that achieves a good approximation of the sequence-to-sequence map of the original state space models of order

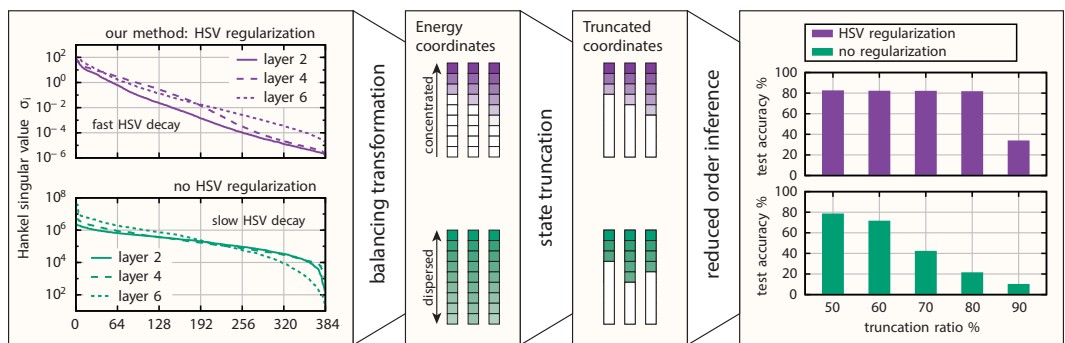

Figure 1: We propose to regularize the Hankel singular values of SSMs so that they become compressible. **Left**: Regularizing with the Hankel singular values during training leads to SSMs with a fast Hankel singular value (HSV) decay. **Middle**: SSMs with a fast HSV decay have many low-energy states that only contribute little to the layer output. **Right**: Compressing the SSM by truncating only such low-energy states changes the corresponding sequence-to-sequence map insignificantly and retains the overall accuracy. Without our regularization, HSVs decay slowly and compression leads to an accuracy deterioration.

$n$. The results in Figure 1 are in agreement with other attempts of compressing SSMs such as [13], which show that on standard LRA benchmarks the Hankel singular values of the SSM layers also decay slowly and thus the systems are not well compressible when trained with default procedures.

## 1.3 Literature review

**Models with state space layers**  DNNs with state space models as layers have recently been made popular starting with the introduction and utilization of the high-order polynomial projection operators (HiPPO) framework in [22], which was applied as *structured state space sequence model* (S4) to efficiently model long sequences in [23] and outperformed the state of the art in the Long Range Arena (LRA) benchmark [59]. This was a major step forward as other architectures such as recurrent neural networks (RNNs) [2, 12, 51, 8] and transformers [60] and memory efficient variants of transformer layers [10, 33, 35, 4, 25, 61] achieved poorer performance at the time on long-range sequence tasks as in [59]. After the initial introduction of S4, a simplified SSM layer (S5) was introduced in [53], in which S4 was streamlined from its original single-input single-output (SISO) and convolutional formulation to a multi-input multi-output (MIMO) time-domain formulation. The time-domain formulation, which leverages a parallel scan for computational efficiency, was an important contribution as it facilitates more variations such as time-varying and input-dependent SSM operators, which are explored independently in *Liquid-S4* [28].

Large scale SSMs are explored in Mamba-S6 [21], which introduces selective SSMs that allow for efficient filtering and context compression, and H3 [15], which is extended further in [49]. The new class of structured SSMs has found a wide range of applications such as audio generation [17] and vision tasks [44, 31].

Even methodological contributions to the layer architecture are still made such as a recent state-free transfer-function-based implementation [46], novel parameterization schemes [63], a reformulation of RNNs in the SSM framework [45], or bidirectional extensions of Mamba [30].

**Compression and distillation of SSMs**  Given that costs and the number of parameters grow with $n$, the order of the SSM, there has been investigations about how to keep $n$ small without sacrificing expressivity. While pruning and post-training compression is extensively studied for transformer architectures [43, 7, 58], model compression for SSM and mixed models has only recently started to gain traction [16, 42, 56]. The work [26] develops system and control-inspired criteria for state-pruning over multiple layers. In contrast, we propose to regularize the Hankel singular values already during training, which is an approach from classical system and control theory for system identification [48, 55, 20] and has been pursued in the context of deep networks with state space model layers in [14]. Our work combines the ideas of [14] and [26]: we leverage Hankel singular value regularization as in [14] and multi-layer state pruning as in [26]. Combining Hankel singular value regularization with multi-layer state pruning enables layer-dependent rank adaptation, which is important when networks get deeper as necessary for the benchmarks we consider. In contrast, the work [14] uses the same rank for all layers. Moreover, we consider a block-structured real-valued parametrization of stable systems, for which we propose novel algorithmic methods for the efficient Hankel singular value computation compared to the diagonal matrices and corresponding algorithms considered in [14]. Combining the layer-adaptive rank, the block-diagonal structure, and our novel algorithms for computing singular values allows us to demonstrate scalability on Long Range Arena benchmarks, thereby highlighting relevance to language modeling tasks beyond the physics problems considered in [14]. Directly applying general model reduction methods to SSMs is attempted in [13] but the method in [13] requires retraining a smaller model after the reduction, which can be a costly step that our approach avoids. Additionally, there is distillation that aims to learn a smaller model from a larger teacher model; see, e.g., [29]. Distillation in the context of SSMs is explored in [43].

## 1.4 Our approach and summary of contributions

We propose a training procedure that allows efficiently regularizing the Hankel singular values of the SSM layers so that the singular values decay favorably for compression. Our contributions are:

- Connecting SSM layers to system theory for deriving conditions under which layers are compressible.

- Regularizing Hankel singular values to nudge the optimizer to seek models that can be well compressed with standard tools from system theory such as balanced truncation, while maintaining accurate sequence-to-sequence mappings. To this end, we prove the differentiability of a nuclear norm regularizer constructed from the Hankel singular values.

- For efficient training, developing an algorithm that enables the computation of so-called gramians, which are needed to evaluate the Hankel singular values of SSMs, with our imposed block structure, that reduces the computational cost from $\mathcal{O}(n^3)$ to $\mathcal{O}(n^2)$ in the state dimension $n$.

- Demonstrating that applying our method can lead to up to a $10\times$ improvement in accuracy for strongly compressed models on Long Range Arena benchmarks.

## 2 Hankel singular value regularization (HSVR)

We propose to regularize the training of neural-network models with SSM layers so that the SSMs have a fast Hankel singular value decay which means they can be compressed efficiently. The key to regularizing the Hankel singular values is ensuring that the regularizer building on them is differentiable and that computing the singular values is efficient during the training iterations, for which we introduce a scalable parametrization and a scalable algorithm.

### 2.1 Parametrization of SSMs for scalable computation of Hankel singular values

**Time-discrete SSMs and their Hankel singular values**   Each SSM layer consists of a linear time-invariant dynamical system,

$$
\begin{aligned}
\boldsymbol{x}_{k+1} &= \boldsymbol{A}\boldsymbol{x}_k + \boldsymbol{B}\boldsymbol{u}_k, \quad \boldsymbol{x}_0 = \boldsymbol{0} \in \mathbb{R}^n, \\
\boldsymbol{y}_k &= \boldsymbol{C}\boldsymbol{x}_k + \boldsymbol{D}\boldsymbol{u}_k,
\end{aligned}
\tag{1}
$$

where $\boldsymbol{A} \in \mathbb{R}^{n \times n}, \boldsymbol{B} \in \mathbb{R}^{n \times p}, \boldsymbol{C} \in \mathbb{R}^{p \times n}$, and $\boldsymbol{D} \in \mathbb{R}^{p \times p}$, are the system, input, output, and feedthrough matrices, respectively, and $\{\boldsymbol{u}_k\}_{k=0}^{n_s}$ and $\{\boldsymbol{y}_k\}_{k=0}^{n_s}$ are the input signal and the system output, respectively. The controllability gramian $\boldsymbol{P}$ and observability gramian $\boldsymbol{Q}$ of size $n \times n$ of the system (1) carry information about the singular values of the underlying Hankel operator $\mathcal{H}$: For stable, controllable, and observable systems (terms defined in the Appendix A), the non-zero Hankel singular values $\sigma_1, \sigma_2, \ldots$ of the system (1) are the square-roots of the eigenvalues of the product $\boldsymbol{PQ}, \sigma_i(\mathcal{H}) = \sqrt{\lambda_i(\boldsymbol{PQ})}$, where $\lambda_i$ denotes the $i$-th eigenvalue of its matrix argument. The gramians can be computed as the solutions to the discrete Lyapunov equations

$$
\boldsymbol{A}\boldsymbol{P}\boldsymbol{A}^\top - \boldsymbol{P} + \boldsymbol{B}\boldsymbol{B}^\top = \boldsymbol{0},
\tag{2}
$$

$$
\boldsymbol{A}^\top\boldsymbol{Q}\boldsymbol{A} - \boldsymbol{Q} + \boldsymbol{C}^\top\boldsymbol{C} = \boldsymbol{0}.
\tag{3}
$$

A straightforward solution to these Lyapunov equations can be computed by vectorizing both equations, which yields

$$
\mathsf{vec}(\boldsymbol{P}) = -(\boldsymbol{A} \otimes \boldsymbol{A} - \boldsymbol{I})^{-1}\mathsf{vec}(\boldsymbol{B}\boldsymbol{B}^\top),
\tag{4}
$$

$$
\mathsf{vec}(\boldsymbol{Q}) = -(\boldsymbol{A}^\top \otimes \boldsymbol{A}^\top - \boldsymbol{I})^{-1}\mathsf{vec}(\boldsymbol{C}^\top\boldsymbol{C}),
\tag{5}
$$

where vec denotes the vectorization operator that stacks all columns of its matrix argument. However, computing $\boldsymbol{P}$ and $\boldsymbol{Q}$ using (4) scales as $\mathcal{O}(n^6)$. The Bartels-Steward algorithm [3] can be used to solve general Lyapunov equations in $\mathcal{O}(n^3)$ using generalized eigen-decompositions.

**Parametrization via rotation matrices**   We now parametrize discrete dynamical systems (1) via scaled rotation matrices. Scaled rotation matrices lead to favorable properties such as making it easy to enforce stability. Furthermore, the rotation block structure allows us to derive algorithms to compute solutions of the Lyapunov equations (2)–(3) with costs that scale as $\mathcal{O}(n^2)$ (instead of $\mathcal{O}(n^3)$ as in the standard Bartels-Steward algorithm). Note that we are not claiming we are the first to recognize the benefits of parametrizing SSMs with rotation matrices as a discrete-time alternative to the HiPPO framework (see, e.g., [11]) but we show how they are useful for efficiently computing Hankel singular values for regularization.

In the following, we set $q = n/2$ and only consider the case $m = p$ to ease the notation burden. At layer $\ell$, we have a system of the form (1) with matrices

$$
\boldsymbol{A}^{(\ell)} = \begin{bmatrix} \boldsymbol{A}_1(\rho_1^{(\ell)}, \alpha_1^{(\ell)}) & 0 & \cdots & 0 \\ 0 & \boldsymbol{A}_2(\rho_2^{(\ell)}, \alpha_2^{(\ell)}) & \cdots & 0 \\ \vdots & \vdots & \ddots & \vdots \\ 0 & 0 & \cdots & \boldsymbol{A}_q(\rho q^{(\ell)}, \alpha_q^{(\ell)}) \end{bmatrix}, \quad \boldsymbol{B}^{(\ell)} = \begin{bmatrix} \boldsymbol{e}_1 & \boldsymbol{B}_1^{(\ell)} \\ \boldsymbol{e}_1 & \boldsymbol{B}_2^{(\ell)} \\ \vdots & \vdots \\ \boldsymbol{e}_1 & \boldsymbol{B}_q^{(\ell)} \end{bmatrix}, \quad (6)
$$

where $\boldsymbol{e}_1 = [1, 0]^\top$ and each block in $\boldsymbol{A}^{(\ell)}$ is a rotation matrix

$$
\boldsymbol{A}_i(\rho_i^{(\ell)}, \alpha_i^{(\ell)}) = \rho_i^{(\ell)} \begin{bmatrix} \cos(\alpha_i^{(\ell)}) & \sin(\alpha_i^{(\ell)}) \\ -\sin(\alpha_i^{(\ell)}) & \cos(\alpha_i^{(\ell)}) \end{bmatrix} \in \mathbb{R}^{2 \times 2}, \quad \boldsymbol{B}_i^{(\ell)} = \begin{bmatrix} \boldsymbol{b}_{i1}^{(\ell)} \\ \boldsymbol{b}_{i2}^{(\ell)} \end{bmatrix} \in \mathbb{R}^{2 \times (p-1)}, \quad (7)
$$

which has the complex conjugate eigenvalues $\rho_i^{(\ell)} \left( \cos(\alpha_i^{(\ell)}) \pm \mathrm{i} \sin(\alpha_i^{(\ell)}) \right)$. The output matrix $\boldsymbol{C}^{(\ell)}$ of size $p \times n$ has no special structure. In our experiments, we only use diagonal feed-through matrices $\boldsymbol{D}^{(\ell)}$ of size $p \times p$; which is standard also in other works [53].

We can summarize the parameters of the SSM in layer $\ell$ in the parameter vector $\boldsymbol{\theta}^{(\ell)} \in \mathbb{R}^{2q + n(p-1) + pn + p}$, which contains in vectorized form $\boldsymbol{\alpha}^{(\ell)}, \boldsymbol{\rho}^{(\ell)}, \boldsymbol{b}_{11}^{(\ell)}, \ldots, \boldsymbol{b}_{q2}^{(\ell)}, \boldsymbol{C}^{(\ell)}, \boldsymbol{D}^{(\ell)}$. Notice that the number of SSM-parameters per layer, i.e., the dimension of $\boldsymbol{\theta}^{(\ell)}$, grows linearly in the order $n$ of the SSM, which is the same as using diagonal system matrices as in the S5 SSM layers [53].

The parametrization given by rotation matrices is universal in the sense that it is dense in the space of linear time-invariant systems of order $n$, which is shown in the following. Recall that systems with the same impulse response are equivalent in the sense that they describe the same sequence-to-sequence map.

**Proposition 1.** *For any linear time-invariant system of order $n$ there exists an infinitesimal perturbation such that the sequence-to-sequence map $\{\boldsymbol{u}_k\}_{k=0}^\infty \mapsto \{\boldsymbol{y}_k\}_{k=0}^\infty$ of the perturbed model can be described by an SSM with matrices of the form (7) with $\boldsymbol{\alpha}, \boldsymbol{\rho} \in \mathbb{R}^{\tilde{n}}$, where $\tilde{n} \le n$.*

Our proof, which we present in the Appendix A.1, uses [34, Chapter 2.2] to break up any Jordan blocks with infinitesimal perturbations, exploits the fact that controllable systems are dense in the space of $n$-dimensional systems [54, Proposition 3.3.12], and finally uses the block Schur decomposition to bring $\boldsymbol{A}$ into the desired form and applies Givens rotations to establish the first column of $\boldsymbol{B}$ as repeated standard-basis vectors.

**Enforcing stability** Stability of SSMs parametrized via rotation matrices can be achieved by enforcing that the entries of the $\boldsymbol{\rho}$ vector remain below one for all layers. Thus, for example, it is sufficient to threshold the entries of $\boldsymbol{\rho}$ with $\tanh$, which we use in our implementation. Moreover, we scale the parameters $\boldsymbol{\alpha}$ using $\tanh$ to be contained in the interval $[0, \pi]$. With this parametrization, we can attain any pair of complex-conjugate eigenvalues inside the unit-disk and thus have a simple and differentiable parametrization of stable systems.

## 2.2 Scalable training procedure

**Leveraging block-diagonal structure for computing singular values** Recall that if we want to regularize the distribution of the Hankel singular values, then we have to compute them at least once in each gradient-descent step during training. The standard algorithm for computing Hankel singular values is solving the Lyapunov equations (2) and (3) with the Bartels-Stewart algorithm [3], which incurs costs that scale as $\mathcal{O}(n^3)$ with the system order $n$. We now introduce an algorithm that leverages the block-diagonal structure (6) of our system parametrization and achieves a cost scaling of $\mathcal{O}(n^2)$. We describe the algorithm for the control Lyapunov equation (2); the treatment of (3) is analogous.

Plugging our parametrization (6) into the control Lyapunov equation (2) leads to

$$
\begin{bmatrix} \boldsymbol{A}_1 & 0 & \cdots & 0 \\ 0 & \boldsymbol{A}_2 & \cdots & 0 \\ \vdots & \vdots & \ddots & \vdots \\ 0 & 0 & \cdots & \boldsymbol{A}_q \end{bmatrix} \begin{bmatrix} \boldsymbol{P}_{11} & \boldsymbol{P}_{12} & \cdots & \boldsymbol{P}_{1q} \\ \boldsymbol{P}_{12}^\top & \boldsymbol{P}_{22} & \cdots & \boldsymbol{P}_{2q} \\ \vdots & \vdots & \ddots & \vdots \\ \boldsymbol{P}_{1q}^\top & \boldsymbol{P}_{2q}^\top & \cdots & \boldsymbol{P}_{qq} \end{bmatrix} \begin{bmatrix} \boldsymbol{A}_1^\top & 0 & \cdots & 0 \\ 0 & \boldsymbol{A}_2^\top & \cdots & 0 \\ \vdots & \vdots & \ddots & \vdots \\ 0 & 0 & \cdots & \boldsymbol{A}_q^\top \end{bmatrix} - \begin{bmatrix} \boldsymbol{P}_{11} & \boldsymbol{P}_{12} & \cdots & \boldsymbol{P}_{1q} \\ \boldsymbol{P}_{12}^\top & \boldsymbol{P}_{22} & \cdots & \boldsymbol{P}_{2q} \\ \vdots & \vdots & \ddots & \vdots \\ \boldsymbol{P}_{1q}^\top & \boldsymbol{P}_{2q}^\top & \cdots & \boldsymbol{P}_{qq} \end{bmatrix} + \begin{bmatrix} \boldsymbol{B}_1 \\ \boldsymbol{B}_2 \\ \vdots \\ \boldsymbol{B}_q \end{bmatrix} \begin{bmatrix} \boldsymbol{B}_1 \\ \boldsymbol{B}_2 \\ \vdots \\ \boldsymbol{B}_q \end{bmatrix}^\top = \boldsymbol{0},
$$

where each $\boldsymbol{P}_{ij} \in \mathbb{R}^{2\times 2}$. This decomposes into $q$ Lyapunov equations

$$\boldsymbol{A}_i \boldsymbol{P}_{ii} \boldsymbol{A}_i^\top - \boldsymbol{P}_{ii} + \boldsymbol{B}_i \boldsymbol{B}_i^\top = \mathbf{0},$$

for $i \in \{1, \ldots, q\}$ and $q(q-1)/2$ Sylvester equations (note that the solution $\boldsymbol{P}$ of (2) is symmetric)

$$\boldsymbol{A}_i \boldsymbol{P}_{ij} \boldsymbol{A}_j^\top - \boldsymbol{P}_{ij} + \boldsymbol{B}_i \boldsymbol{B}_j^\top = \mathbf{0}, \tag{8}$$

for $i \in \{1, \ldots, q\}$ and $j \in \{i+1, \ldots, q\}$, which can be solved similarly by vectorizing as in (4), which is efficient for small 2-by-2 blocks. As each block $\boldsymbol{P}_{ij}$ can be solved independently, solving for $\boldsymbol{P}$ can be efficiently parallelized. The overall costs scale as $\mathcal{O}(q^2)$ and thus as $\mathcal{O}(n^2)$ because $q = n/2$. Note that the computational costs of the regularizer is independent of the sequence length $n_s$.

After computing the gramians $\boldsymbol{P}$ and $\boldsymbol{Q}$, the Hankel singular values are the singular values of the product $\boldsymbol{PQ}$, which can be computed using standard algorithms. Note that the block structure in the system matrix $\boldsymbol{A}$ does not imply a block structure in $\boldsymbol{P}$ and $\boldsymbol{Q}$. This is again because the Hankel operator does not consider the subsystems separately but considers their interaction as well. For a block structure in $\boldsymbol{P}$ and $\boldsymbol{Q}$, the matrices $\boldsymbol{B}$ and $\boldsymbol{C}$ would have to be block-diagonal as well, which limits the expressivity of the SSM, and thus of the corresponding neural-network model.

**Fast time integration with associative scan**    A major ingredient for making SSMs scalable is using parallel scans for computing output sequences. For diagonal matrices $\boldsymbol{A}$, a parallel scan version has been introduced in [53], but it critically depends on $\boldsymbol{A}$ to be a diagonal matrix to keep the costs in $\mathcal{O}(n)$. The work [11] builds on rotation matrices as well and proposes a parallel scan version that explicitly calculates products of the blocks. In contrast, we now show that an analogous associative scan operation exists for our parametrization via rotation matrices that avoids having to compute products of the blocks explicitly.

For an associative binary operator $\star$, i.e. an operator such that $(a \star b) \star c = a \star (b \star c)$, and a sequence of elements $[a_1, \ldots, a_{n_s}]$, the scan operation returns $[a_1, (a_1 \star a_2), \ldots, (a_1 \star a_2 \star \cdots \star a_{n_s})]$. In [53], the authors consider the associative binary operation

$$\star : (\mathbb{C}^{n\times n}, \mathbb{C}^n), (\mathbb{C}^{n\times n}, \mathbb{C}^n) \to (\mathbb{C}^{n\times n}, \mathbb{C}^n), ((\mathbf{X}, \mathbf{x}), (\boldsymbol{Y}, \boldsymbol{y})) \mapsto (\boldsymbol{Y}\boldsymbol{X}, \boldsymbol{Y}\boldsymbol{x} + \boldsymbol{y}),$$

where the matrix-matrix product $\boldsymbol{Y}\boldsymbol{X}$ can be computed in $\mathcal{O}(n)$, when $\boldsymbol{X}$ and $\boldsymbol{Y}$ are diagonal matrices. This is the key why SSMs with diagonal matrices in [53] are scalable. The sequence, on which the scan operates, is then initialized as

$$a = [(\boldsymbol{A}, \boldsymbol{B}\boldsymbol{u}_1), (\boldsymbol{A}, \boldsymbol{B}\boldsymbol{u}_2), \ldots, (\boldsymbol{A}, \boldsymbol{B}\boldsymbol{u}_{n_s})]. \tag{9}$$

It is easy to verify that the scan with $\star$ over $a$ leads to the state sequence of (1) by noting that the scan output elements can be written as $s_i = (\boldsymbol{A}^i, \sum_{k=1}^{i} \boldsymbol{A}^{i-k} \boldsymbol{B}\boldsymbol{u}_k)$. The second element of each tuple $s_i$ is exactly the state $\boldsymbol{x}_i$ for a discrete system (1). Here $\boldsymbol{A}^i$ denotes the $i$-fold matrix product of $\boldsymbol{A}$ with itself.

For our rotation-based parametrization, we can define a similar associative binary operation

$$\tilde{\star} : (\mathbb{R}^q, \mathbb{R}^q, \mathbb{R}^n), (\mathbb{R}^q, \mathbb{R}^q, \mathbb{R}^n) \to (\mathbb{R}^q, \mathbb{R}^q, \mathbb{R}^n), \tag{10}$$

$$(\boldsymbol{x}^{(1)}, \boldsymbol{x}^{(2)}, \boldsymbol{x}^{(3)}), (\boldsymbol{y}^{(1)}, \boldsymbol{y}^{(2)}, \boldsymbol{y}^{(3)}) \mapsto ((\boldsymbol{x}^{(1)} \odot \boldsymbol{y}^{(1)}), (\boldsymbol{x}^{(2)} + \boldsymbol{y}^{(2)}), \boldsymbol{A}(\boldsymbol{y}^{(1)}, \boldsymbol{y}^{(2)}) \boldsymbol{x}^{(3)} + \boldsymbol{y}^{(3)}),$$

where $\odot$ denotes the Hadamard product and $\boldsymbol{A}(\cdot, \cdot)$ is formed as in (6) for its vector-valued arguments. We can then initialize the scan sequence for each layer with

$$b = [(\boldsymbol{\rho}, \boldsymbol{\alpha}, \boldsymbol{B}\boldsymbol{u}_1), (\boldsymbol{\rho}, \boldsymbol{\alpha}, \boldsymbol{B}\boldsymbol{u}_2), \ldots, (\boldsymbol{\rho}, \boldsymbol{\alpha}, \boldsymbol{B}\boldsymbol{u}_{n_s})].$$

We can verify that a scan with $\tilde{\star}$ over $b$ leads to the state sequence because the scan output elements can be written as $s_i = (\boldsymbol{A}(\boldsymbol{\rho}, \boldsymbol{\alpha}))^i, \sum_{k=1}^{i} (\boldsymbol{A}(\boldsymbol{\rho}, \boldsymbol{\alpha}))^{i-k} \boldsymbol{B}\boldsymbol{u}_k)$. Here we use the fact that for scalars $x, y, \beta, \gamma$, we have that

$$x \begin{bmatrix} \cos(\beta) & \sin(\beta) \\ -\sin(\beta) & \cos(\beta) \end{bmatrix} y \begin{bmatrix} \cos(\gamma) & \sin(\gamma) \\ -\sin(\gamma) & \cos(\gamma) \end{bmatrix} = xy \begin{bmatrix} \cos(\beta + \gamma) & \sin(\beta + \gamma) \\ -\sin(\beta + \gamma) & \cos(\beta + \gamma) \end{bmatrix},$$

such that we can use that $\boldsymbol{A}(\boldsymbol{x}, \boldsymbol{\beta}) \boldsymbol{A}(\boldsymbol{y}, \boldsymbol{\gamma}) = \boldsymbol{A}(\boldsymbol{x} \odot \boldsymbol{y}, \boldsymbol{\beta} + \boldsymbol{\gamma})$ in (10). This avoids having to compute the product of all blocks on the diagonal as in [11]. Using our parallel scan operation, the costs of generating an output sequence of length $n_s$ scale as $\mathcal{O}(\log(n_s)n)$, assuming $n_s$ processors which is the same scaling as when using diagonal matrices as in [53].

## 2.3 Regularizing Hankel singular values during training

**Differentiable regularizers involving Hankel singular values**  Building on the efficient computation of Hankel singular values from the previous section, we now develop a regularizer $\mathcal{R}$ that depends on the Hankel singular values of all layers. We stress that building regularizers based on the Hankel singular values is standard practice in systems and control theory [48, 55, 20] and has been proposed in the work [14] for deep state-space models for the first time; see Section 1.3 for an in-depth comparison to these works.

First, we show a new result that even though the individual Hankel singular values are not differentiable with respect to the entries of the system matrices (6) (i.e., the network parameters), the sum of the Hankel singular values is differentiable. We follow similar arguments as used in [52, Proposition 3.7], which uses that different branches of singular value curves that intersect each other and form a non-simple singular value still add up smoothly locally. Denote with $\boldsymbol{\sigma}^{(\ell)} = [\sigma_1^{(\ell)}, \dots, \sigma_n^{(\ell)}]$ the singular values of the system at layer $\ell$.

**Proposition 2.** *Given an asymptotically stable matrix $\boldsymbol{A}$, as well as $\boldsymbol{B}$ and $\boldsymbol{C}$ such that the pairs $(\boldsymbol{A}, \boldsymbol{B})$ and $(\boldsymbol{A}, \boldsymbol{C})$ are controllable and observable, respectively, let $\boldsymbol{P}$ and $\boldsymbol{Q}$ be the solutions (2) and (3), respectively. Then the sum of Hankel singular values $\sum_{i=1}^{n} \sigma_i$ of $\boldsymbol{P}\boldsymbol{Q}$ depends smoothly on $\boldsymbol{A}$, $\boldsymbol{B}$, and $\boldsymbol{C}$.*

For a proof, see Appendix A.2.

**Regularizing the Hankel nuclear norm**  Nuclear norm regularization, i.e., penalizing the sum of singular values of a matrix, to encourage singular values to decay rapidly, is common practice in machine learning; see, e.g., [20, 14] for uses cases in reducing systems. With the tools we just developed, we can now regularize sums of Hankel singular values of SSM layers for a fast Hankel singular value decay. In particular, the Hankel nuclear norm of a system is the sum of its Hankel singular values.

For this, we introduce the regularizer

$$\mathcal{R}_*(\boldsymbol{\sigma}^{(1)}, \dots, \boldsymbol{\sigma}^{(L)}) = \sum_{\ell=1}^{L} \sum_{i=1}^{n} \sigma_i^{(\ell)} \,, \tag{11}$$

which is added to the loss during training. Recall that Proposition 2 in combination with our parametrization guarantees that $\mathcal{R}_*$ is differentiable with respect to the neural-network parameters.

## 2.4 Compressing the trained models (post-processing)

**Compressing regularized SSMs with balanced truncation**  After having trained a DNN with SSM layers with regularized Hankel singular values, we can apply off-the-shelf model reduction methods to compress the SSM layers. We use balanced truncation to compute the compressed (reduced) systems, because it is well studied and developed and the reduced system inherits favorable properties such as stability from the original, full system. We use the standard square-root method [6, Chapter 6.2] to compute the balanced truncation SSM with reduced state dimension. For this, in each layer we first compute the final controllability and observability gramians $\boldsymbol{P}$ and $\boldsymbol{Q}$ by solving (2) and (3). After that, we compute the singular value decomposition $\boldsymbol{\Phi}\boldsymbol{\Sigma}\boldsymbol{\Psi}^\top = \mathsf{svd}(\boldsymbol{S}^\top \boldsymbol{R})$, where $\boldsymbol{S}\boldsymbol{S}^\top = \boldsymbol{Q}$ and $\boldsymbol{R}\boldsymbol{R}^\top = \boldsymbol{P}$ are Cholesky decompositions of $\boldsymbol{Q}$ and $\boldsymbol{P}$. Then we can define the projection matrices $\boldsymbol{V} = \boldsymbol{S}^\top \boldsymbol{\Phi}_{:,:r} \boldsymbol{\Sigma}_{:r,:r}^{-\frac{1}{2}}$ and $\boldsymbol{W} = \boldsymbol{R}^\top \boldsymbol{\Psi}_{:,:r} \boldsymbol{\Sigma}_{:r,:r}^{-\frac{1}{2}}$, where for a matrix $\boldsymbol{M}$, the expression $\boldsymbol{M}_{:,:r}$ denotes the first $r$ columns of $\boldsymbol{M}$ and $\boldsymbol{M}_{:r,:r}$ denotes the upper left $r$-dimensional block of $\boldsymbol{M}$. The reduced $r$-dimensional SSM is then obtained as $[\boldsymbol{W}^\top \boldsymbol{A}\boldsymbol{V}, \boldsymbol{W}^\top \boldsymbol{B}, \boldsymbol{C}\boldsymbol{V}, \boldsymbol{D}]$.

For balanced truncation, the error incurred in the sequence-to-sequence map $\{\boldsymbol{u}_k\}_k \mapsto \{\hat{\boldsymbol{y}}_k\}_k$ of the compressed system is bounded as

$$\|\hat{\boldsymbol{y}}_k - \boldsymbol{y}_k\|_{\ell_2} \leq 2\|\boldsymbol{u}\|_{\ell_2} \sum_{i=r+1}^{n} \sigma_i \tag{12}$$

and thus controlled by the sum of truncated Hankel singular values. Thus, the bound (12) provides a viable criterion for choosing the compression order $r$ in the different layers; see next paragraph.

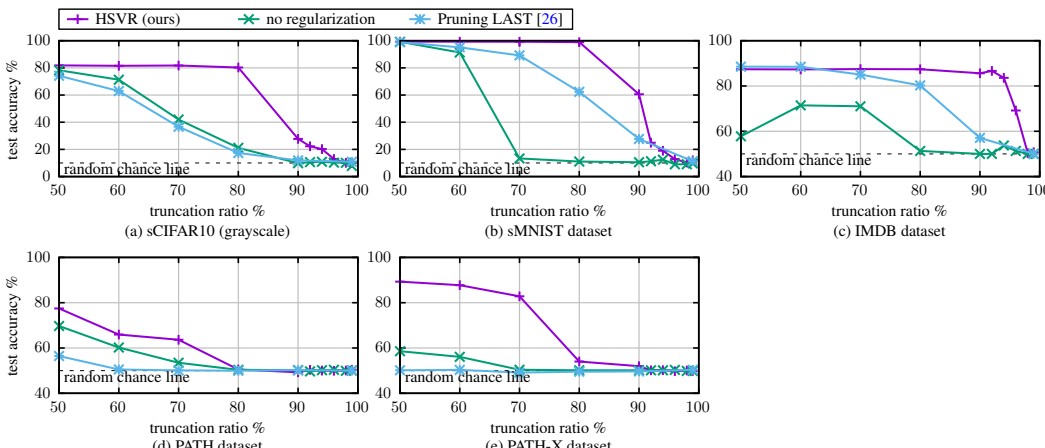

Figure 2: Regularizing Hankel singular values leads to highly compressible SSMs while maintaining accuracy.

**Balancing the reduced order $r$ of compressed SSMs across all layers** The Hankel singular values give us guidance to which order $r \ll n$ to compress the regularized SSMs: We define the energy of the SSM at layer $\ell$ as $e^{(\ell)} = \sum_{i=1}^{n} \sigma_i^{\ell}$, which is the sum of all Hankel singular values of the SSM at layer $\ell$. We then prescribe a criterion such as retaining 99% of all energy, which means truncating at order $r$ so that $e_r^{\ell}/e^{\ell} = 0.99$ for $e_r^{\ell} = \sum_{i=1}^{r} \sigma_i^{(\ell)}$. We stress that there is a one-to-one correspondence to the error incurred in the sequence-to-sequence map, because of the bound (12) satisfied by models compressed with balanced truncation. Notice that we prescribe the same energy criterion (e.g., 99%) for all layers $\ell = 1, \ldots, L$ but that the corresponding compression order $r_1, \ldots, r_L$ can be different for each layer.

Alternatively, we can prescribe a total budget $r_t = \sum_{\ell=1}^{L} r^{(\ell)}$ of state dimensions, where $r^{(1)}, \ldots, r^{(\ell)}$ are the state dimensions of the SSMs corresponding to layers $\ell = 1, \ldots, L$. Given a total budget $r_t$, we can then distribute the state dimensions across the $\ell = 1, \ldots, L$ layers such that the same amount of energy is preserved in each layer. For this, we use a bisection algorithm that is described in detail in the Appendix in Section C.1.

**Diagonalizing compressed systems** The reduced systems are balanced but not necessarily diagonal or block-diagonal, which is essential for an efficient application of the associative scan operations; see Section 2.2. We diagonalize the compressed SSMs using an eigenvalue decomposition: Let $\boldsymbol{A}_r = \boldsymbol{W}^{\top}\boldsymbol{A}\boldsymbol{V}$ be the reduced system matrix. Then we can compute an eigenvalue decomposition $\boldsymbol{T}\boldsymbol{\Lambda}_r\boldsymbol{T}^{-1} = \boldsymbol{A}_r$, where $\boldsymbol{\Lambda}_r \in \mathbb{R}^{r \times r}$ is diagonal. An equivalent diagonal system to $[\boldsymbol{W}^{\top}\boldsymbol{A}\boldsymbol{V}, \boldsymbol{W}^{\top}\boldsymbol{B}, \boldsymbol{C}\boldsymbol{V}, \boldsymbol{D}]$ is then given by $[\boldsymbol{T}^{-1}\boldsymbol{W}^{\top}\boldsymbol{A}\boldsymbol{V}\boldsymbol{T}, \boldsymbol{T}^{-1}\boldsymbol{W}^{\top}\boldsymbol{B}, \boldsymbol{C}\boldsymbol{V}\boldsymbol{T}, \boldsymbol{D}]$. The diagonalized system might be complex-valued like the systems in [53]; however, since the eigenvalues will appear in complex-conjugate pairs, a real-valued input sequence will be mapped to a real-valued output sequence, which is also used in [53].

## 3 Results

**Benchmarks** We demonstrate our HSVR approach on five sequence classification examples. The first example consists of the 32×32 CIFAR-10 images [37] that are converted to grayscale, flattened into 1,024-length sequences, and normalized to zero mean and unit variance across the entire dataset. It includes 50,000 training, and 10,000 test samples and has ten target classes. The second example is also a sequentialized image classification task and consists of the 28×28 grayscale MNIST [38] images, where again each image is flattened into a sequence of 784 scalar values. The goal is to predict the depicted written digits correctly. The third task uses the IMDB sentiment dataset [41], where movie reviews are represented as sequences of one-hot encoded characters with 129 possible values, padded to a maximum length of 4,096. The goal is to classify each review as positive or negative; the dataset includes 25,000 training and 25,000 test examples. Finally, we consider the

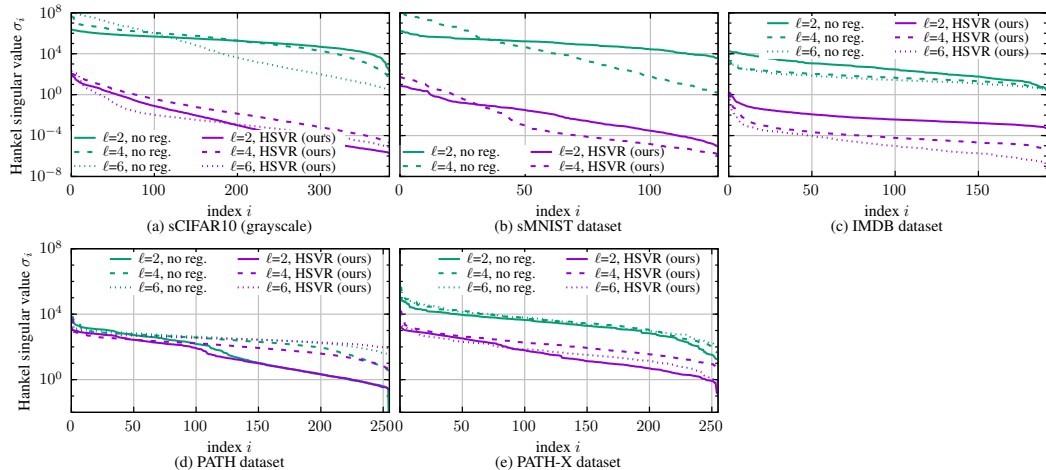

Figure 3: Our HSVR approach trains SSMs that have favorably Hankel singular value decay for compression.

Table 1: Test accuracies for different methods for different truncation ratios.

| Method | sCIFAR (grayscale) | | | | sMNIST | | | | IMDB | | | |
|---|---|---|---|---|---|---|---|---|---|---|---|---|
| trunc. ratio | 60% | 70% | 80% | 90% | 60% | 70% | 80% | 90% | 60% | 70% | 80% | 90% |
| LAST [26] | 62.93 | 36.66 | 17.35 | 11.19 | 95.11 | 89.17 | 62.37 | 27.67 | **88.48** | 85.05 | 80.26 | 57.08 |
| global [26] | 28.91 | 13.62 | 11.12 | 10.47 | 91.67 | 83.32 | 52.52 | 21.94 | 88.28 | **87.70** | 83.75 | 63.80 |
| uniform [26] | 58.90 | 34.45 | 19.18 | 12.67 | 97.74 | 79.20 | 44.38 | 23.20 | 82.44 | 77.34 | 64.79 | 53.22 |
| no reg. | 71.28 | 41.98 | 21.14 | 9.84 | 91.32 | 13.35 | 11.05 | 10.55 | 71.45 | 71.04 | 51.32 | 50.00 |
| HSVR (ours) | **81.84** | **81.75** | **81.37** | **51.08** | **99.45** | **99.22** | **98.90** | **86.95** | 87.26 | 87.16 | **86.97** | **86.40** |

| Method | PATH | | | | PATH-X | | | |
|---|---|---|---|---|---|---|---|---|
| trunc. ratio | 60% | 70% | 80% | 90% | 60% | 70% | 80% | 90% |
| LAST [26] | 50.51 | 50.11 | 49.96 | **50.25** | 50.33 | 49.16 | 49.53 | 49.64 |
| global [26] | 49.16 | 50.15 | 50.16 | 49.37 | 49.50 | 50.93 | 49.14 | 50.61 |
| uniform [26] | 50.32 | 49.78 | 49.84 | 50.16 | 49.74 | 50.23 | 49.70 | 50.47 |
| no reg. | 60.21 | 53.48 | 50.35 | 50.20 | 56.09 | 50.39 | 50.16 | 50.14 |
| HSVR (ours) | **65.94** | **63.64** | **50.50** | 49.27 | **87.74** | **82.82** | **54.02** | **51.97** |

PATH and PATH-X datasets, which consist of the flattened pathfinder images [40], which consists of two points and a set of paths. The classifier must determine whether the two points are connected by the paths. The flattened PATH images have a sequence length of 1,024 and flattened PATH-X images have a sequence length of 16,384a. We denote the examples by sCIFAR (grayscale), sMNIST, IMDB, PATH, and PATH-X. The examples sCIFAR (grayscale), IMDB, PATH, and PATH-X are also part of the Long Range Arena (LRA) benchmark [59] collection.

**Setup**  We select the state, input, and output dimensions of our SSMs according to the setup in [53]. In particular, we use a state dimension $n = 384$, and input and output dimensions $m = p = 512$ for sCIFAR 10 (grayscale), $n = m = p = 128$ for sMNIST, $n = 192, m = p = 256$ for IMDB, $n = 256, m = p = 192$ for PATH, and $n = 256, m = p = 128$ for PATH-X. As in [53] for sCIFAR 10 (grayscale) IMDB, PATH, and PATH-X, we use 6 SSM layers and for sMNIST we use 4 layers. The remainder of the model architecture, which we describe alongside the training parameters in the Appendix in Section B, is also the same as in [53]. In all examples, we use HSVR with the Hankel nuclear norm regularizer (11), even though other regularizers based on the Hankel singular values could be used, which remains future work. One notable difference compared to [53, 26] is that we only use unidirectional associative scans, whereas [53, 26] scan bidirectionally for sCIFAR (grayscale) and IMDB. This means that they apply the associative scan to the given and the reversed sequence and double the dimension of the output matrices of the SSMs to merge both sequences into one output sequence.

**SSMs trained with our HSVR are highly compressible**  In Figure 2, we show the model accuracy on the given test data, as we increase the *truncation ratio* $\chi$, which is the maximum allowed average reduced state dimension across all layers. For an original state dimension $n$, the maximum allowed average reduced state dimension $r$ is $n(1-\chi)$. For our comparison to [26], we extracted the accuracies reported in Figures 2 and 6 in [26]. For HSVR, we show the median over three training runs initialized with different random seeds; standard deviations are reported in the Appendix in Section D. The results in Figure 2 clearly demonstrate the effectiveness of our proposed HSVR. The accuracy of the full SSM model is retained for truncation ratios of $80\%$ for sCIFAR (grayscale) and sMNIST and even for over $90\%$ truncation ratio for the IMDB dataset. Even for the more challenging PATH and PATH-X datasets, we can observe a higher test accuracy for larger truncation ratios compared to other methods. Without regularized training, the accuracy drops much earlier, which is also the case for LAST-based pruning of [26], which also follows the S5 architecture in its experiment setup. The results in Figure 3 provide further evidence that HSVR leads to favorable Hankel singular value decay compared to unregularized training. Note that only for the PATH dataset our HSVR regularizer did not lead to a significant difference in the HSV distribution when comparing with unregularized training; especially when comparing with the HSV distributions for the other datasets in Figure 3, where a clear gap appears between regularized and unregularized training. Moreover, on the PATH dataset, we achieved the smallest test accuracy improvement. This again emphasizes that the HSV distribution is key when considering the compressibility of SSMs.

**HSVR achieves higher compression than previous methods**  In Table 1, we conduct a comparison to all pruning methods proposed in [26] as well as training our models with no regularization. Overall, Table 1 again demonstrates the benefits of HSV regularized training. Our HSVR approach outperforms all other compression methods over a wide range of compression ratios and accuracy ranges; the only exception being at very low compression ratios, which are of less interest in most cases. For example, we maintain accuracy of around to 99% with a compression ratio of 80% in the sMNIST data set, while compressing unregularized SSMs leads to an accuracy drop to almost 10%. Notably, with our regularization we can maintain a high accuracy on the challenging PATH-X dataset even at compression ratios above 60%, where prior methods collapse to random-chance performance.

## 4  Conclusions, limitations, and impact statement

**Conclusions**  We demonstrated that regularizing the Hankel singular values of the SSMs is key for compression. While the individual Hankel singular values are not differentiable, their sum is, which is all that is needed for obtaining a differentiable regularizer. A key aspect is that we developed an algorithm that can efficiently compute the Hankel singular values to keep training costs low. Experiments with standard LRA benchmark examples demonstrate that we can compress models by up to 90% while maintaining acceptable accuracy. An implementation is provided at https://github.com/Algopaul/hankelreg.

**Limitations**  (a) Models need to be trained with the regularizer to achieve compressibility, which means that our compression approach is not applicable to pre-trained models without our regularizer. Because it is known that linear equivalence transformations cannot change the Hankel singular values, it remains future work to find nonlinear transformations to achieve compressibility also for pre-trained models. (b) By regularizing the Hankel singular values and compressing with system-theoretic tools such balanced truncation, we are restricted to linear compression. This is reasonable as the SSMs are linear in the state too but there can exist more efficient nonlinear compressions. Rigorous nonlinear compressions for dynamical systems are an active research direction in systems and control theory [47] and it remains future work to develop corresponding regularizers for SSM layers and neural-network models. (c) We focus on SSMs that are linear time-invariant systems; however, using time-varying system matrices can increase expressivity of the corresponding neural-network models without increasing parameter count and they are explored in MAMBA architecture [21]. It remains future work to extend our approach to systems with time-varying system matrices.

**Impact statement**  We are not expecting negative societal impacts that are specific to our compression approach.

## Acknowledgements

The authors have been partially funded by the Air Force Office of Scientific Research (AFOSR), USA, award FA9550-24-1-0327.

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

## A  Proofs

For our proofs, the following notions for LTI systems are helpful. An LTI system is controllable if the *controllability matrix* $\mathcal{C} = \begin{bmatrix} \boldsymbol{B} & \boldsymbol{AB} & \boldsymbol{A}^2\boldsymbol{B} & \cdots & \boldsymbol{A}^{n-1}\boldsymbol{B} \end{bmatrix}$, i.e. $\operatorname{rank}(\mathcal{C}) = n$. Moreover, the system is *observable* if the observability matrix $\mathcal{O} = \begin{bmatrix} \boldsymbol{C}^\top & (\boldsymbol{CA})^\top & \cdots & (\boldsymbol{CA}^{n-1})^\top \end{bmatrix}^\top$ has full rank, i.e. $\operatorname{rank}(\mathcal{O}) = n$.

The system is *asymptotically stable* if all eigenvalues of $\boldsymbol{A}$ lie strictly inside the unit circle.

### A.1  Parametrization with rotation matrices is dense

We present a proof for Proposition (1), which is restated here for convenience.

**Proposition.** *For any linear time-invariant system of order $n$ there exists an infinitesimal perturbation such that the sequence-to-sequence map $\{\boldsymbol{u}_k\}_{k=0}^\infty \mapsto \{\boldsymbol{y}_k\}_{k=0}^\infty$ of the perturbed model can be described by an SSM with matrices of the form (7) with $\boldsymbol{\alpha}, \boldsymbol{\rho} \in \mathbb{R}^{\tilde{n}}$, where $\tilde{n} \leq n$.*

*Proof.* Let the LTI system be given by $[\boldsymbol{A}, \boldsymbol{B}, \boldsymbol{C}, \boldsymbol{D}]$. As we allow for infinitesimal perturbations, we can assume without loss of generality (w.l.o.g), that $\boldsymbol{A}$ is diagonalizable [34, Chapter 2] and that the pair $(\boldsymbol{A}, \boldsymbol{B})$ is controllable [54, Proposition 3.3.12]. For any nonsingular $\boldsymbol{T} \in \mathbb{R}^{n \times n}$, the impulse response of $[\boldsymbol{A}, \boldsymbol{B}, \boldsymbol{C}, \boldsymbol{D}]$ and $[\boldsymbol{T}^{-1}\boldsymbol{A}\boldsymbol{T}, \boldsymbol{T}^{-1}\boldsymbol{B}, \boldsymbol{C}\boldsymbol{T}, \boldsymbol{D}]$ coincide. We will now construct an invertible $\boldsymbol{T}$ that brings $[\boldsymbol{A}, \boldsymbol{B}, \boldsymbol{C}, \boldsymbol{D}]$ into a form attainable by (7). By [18, Theorem 7.4.1] we can find an orthogonal $\boldsymbol{T}_1$ such that $\boldsymbol{T}_1^{\top}\boldsymbol{A}\boldsymbol{T}$ is in block upper triangular form with either 1-by-1 or 2-by-2 blocks on the diagonal, which contain the real and complex conjugate eigenvalues of $\boldsymbol{A}$, respectively. Moreover, by [18, Theorem 7.1.6] and since we assume diagonalizabiliy, we can find a nonsingular $\boldsymbol{T}_2$ such that $(\boldsymbol{T}_1\boldsymbol{T}_2)^{-1}\boldsymbol{A}(\boldsymbol{T}_1\boldsymbol{T}_2)$ is block diagonal. W.l.o.g, we can assume that each 2-by-2 block has the form

$$T_{ii} = \begin{bmatrix} a_i & b_i \\ -b_i & a_i \end{bmatrix},$$

as the 2-by-2 blocks contain complex conjugate eigenvalues. The final equivalence transformation we apply is a block diagonal transformation that brings the first column of $(\boldsymbol{T}_1\boldsymbol{T}_2)^{-1}\boldsymbol{B}$ into the desired form. For this, let

$$(\boldsymbol{T}_1\boldsymbol{T}_2)^{-1}\boldsymbol{B} = \begin{bmatrix} \boldsymbol{b}_{1,1} & \boldsymbol{b}_{1,2} \\ \boldsymbol{b}_{2,1} & \boldsymbol{b}_{2,2} \\ \vdots & \vdots \\ \boldsymbol{b}_{q,1} & \boldsymbol{b}_{q,2} \end{bmatrix},$$

where $q$ is the number of blocks in $(\boldsymbol{T}_1\boldsymbol{T}_2)^{-1}\boldsymbol{A}(\boldsymbol{T}_1\boldsymbol{T}_2)$ and $\boldsymbol{b}_{i,1}$ is either in $\mathbb{R}^1$ or $\mathbb{R}^2$ depending on the corresponding block dimension in $(\boldsymbol{T}_1\boldsymbol{T}_2)^{-1}\boldsymbol{A}(\boldsymbol{T}_1\boldsymbol{T}_2)$. Now we construct $\boldsymbol{T}_3$ as block diagonal matrix such that $\boldsymbol{T}_3^{-1}$ contains either $1/\boldsymbol{b}_{i,1}$ when $\boldsymbol{b}_{i,1} \in \mathbb{R}^1$ or a scaled Givens rotation $\gamma\boldsymbol{G}$ when $\boldsymbol{b}_{i,1} \in \mathbb{R}^2$ that is constructed such that $\gamma\boldsymbol{G}\boldsymbol{b}_{i,1} = [1, 0]^{\top}$. Note that we can construct such a rotation for $\boldsymbol{b}_{i,1} \in \mathbb{R}^2$ or have a finite value $1/\boldsymbol{b}_{i,1}$ for $\boldsymbol{b}_{i,1} \in \mathbb{R}^1$ since we have assumed controllability. Moreover, we have that $(\boldsymbol{T}_1\boldsymbol{T}_2\boldsymbol{T}_3)^{-1}\boldsymbol{A}(\boldsymbol{T}_1\boldsymbol{T}_2\boldsymbol{T}_3) = (\boldsymbol{T}_1\boldsymbol{T}_2)^{-1}\boldsymbol{A}(\boldsymbol{T}_1\boldsymbol{T}_2)$. W.l.o.g. let us assume for convenience that the blocks in $(\boldsymbol{T}_1\boldsymbol{T}_2)^{-1}\boldsymbol{A}(\boldsymbol{T}_1\boldsymbol{T}_2)$ are sorted such that a partitioning

$$\begin{bmatrix} \boldsymbol{A}_1 & \boldsymbol{0} \\ \boldsymbol{0} & \boldsymbol{A}_2 \end{bmatrix} = (\boldsymbol{T}_1\boldsymbol{T}_2)^{-1}\boldsymbol{A}(\boldsymbol{T}_1\boldsymbol{T}_2)$$

can be chosen such that $\boldsymbol{A}_1$ is diagonal and $\boldsymbol{A}_2$ contains all 2-by-2 blocks. Thus, the system $[\boldsymbol{T}^{-1}\boldsymbol{A}\boldsymbol{T}, \boldsymbol{T}^{-1}\boldsymbol{B}, \boldsymbol{C}\boldsymbol{T}, \boldsymbol{D}]$ with $\boldsymbol{T} = \boldsymbol{T}_1\boldsymbol{T}_2\boldsymbol{T}_3$ where $\boldsymbol{T}^{-1}\boldsymbol{B}$ is of the form

$$\boldsymbol{T}^{-1}\boldsymbol{B} = \begin{bmatrix} 1 & \tilde{\boldsymbol{b}}_{1,2} \\ \vdots & \vdots \\ 1 & \tilde{\boldsymbol{b}}_{q_1,2} \\ \hline 1 & \tilde{\boldsymbol{b}}_{q_1+1,2} \\ 0 & \tilde{\boldsymbol{b}}_{q_1+2,2} \\ 1 & \tilde{\boldsymbol{b}}_{q_1+3,2} \\ 0 & \tilde{\boldsymbol{b}}_{q_1+4,2} \\ \vdots & \vdots \\ 1 & \tilde{\boldsymbol{b}}_{q_1+2q_2,2} \\ 0 & \tilde{\boldsymbol{b}}_{q_1+2q_2,2} \end{bmatrix},$$

where $\tilde{\boldsymbol{b}}_{i,2} \in \mathbb{R}^{1,m}$ and $q_1$ and $q_2$ are the number of real and the number of pairs of complex-conjugate eigenvalues in $\boldsymbol{A}$, respectively. The final step to reach the form (7) is to add a zero row in $\boldsymbol{T}^{-1}\boldsymbol{B}$ after each row in $\{1, \ldots, q_1\}$ and a zero column in $\boldsymbol{C}\boldsymbol{T}$ after each column in $\{1, \ldots q_1\}$. The system matrix is formed from $\tilde{\boldsymbol{A}} = \boldsymbol{T}^{-1}\boldsymbol{A}\boldsymbol{T}$ by choosing $\rho_i = \tilde{\boldsymbol{A}}_i$ and $\alpha_i = 0$, when $\boldsymbol{A}_i$ is a 1-by-1 block and by setting $\rho_i = |\lambda_i|$, $\alpha_i = \arctan(|\Im(\lambda_i)|/\Re(\lambda_i)|)$, when $\boldsymbol{A}_i$ is a 2-by-2 block. Here $\lambda_i$ denotes one of the complex conjugate eigenvalues of $\boldsymbol{A}_i$. Since we only applied equivalence transformations and the states we add in our final step do not change the impulse response (they are uncontrollable and unobservable) the impulse response remains the same. Moreover, since the number of blocks in $\tilde{\boldsymbol{A}}$ is less than or equal to $n$, we can parametrize the system with $\boldsymbol{\rho} = [\rho_1, \ldots, \rho_{\tilde{n}}]$ and $\boldsymbol{\alpha} = [\alpha_1, \ldots, \alpha_{\tilde{n}}]$ where $\tilde{n} \leq n$. $\qquad\square$

### A.2 Proof that sums of Hankel singular values are differentiable

We present a proof for Proposition (2), which is restated here for convenience.

**Proposition.** *Given an asymptotically stable matrix $A$, as well as $B$ and $C$ such that the pairs $(A, B)$ and $(A, C)$ are controllable and observable, respectively, let $P$ and $Q$ be the solutions (2) and (3), respectively. Then the sum of Hankel singular values $\sum_{i=1}^{n} \sigma_i$ depends smoothly on $A$, $B$, and $C$.*

*Proof.* • Note that if $A$ is asymptotically stable, the matrix $(A \otimes A - I)$ has no zero eigenvalues, such that the inverse in (4) exists and $P$ depends smoothly on $A$ and $B$.

- An analogous argument establishes the smooth dependency of $Q$ on $A$ and $C$.

- If all eigenvalues of $PQ$ are simple, the result is automatic.

- For non simple eigenvalues of $PQ$, we can use [34, Theorem 6.8] to define an set of smooth functions representing the repeated eigenvalues, such that their sum is differentiable as well. Note that controllability and observability of $A, B, C$ ensures that $PQ$ has no zero eigenvalues. $\qquad\square$

## B   Experimental setup

For the model architecture, we closely follow the setup proposed in [53] for comparability. In particular, we use the same number of layers and SSM dimensions. Moreover, as in [53], the SSM layers in between linear encoder and decoder layers, that map the dimension of the sequence (1 in all our experiments) to the SSM input dimension, and the SSM output dimension to the number of classes for classification, respectively. In after each sequence layer, we apply the same nonlinearity as [53], which is a weighted sigmoid gated unit [57]. It transforms the SSM output $y_t$ such that $\tilde{y}_t = \mathsf{gelu}(y_t) \odot \mathsf{sigmoid}(W\mathsf{gelu}(y_t))$, with a learnable matrix $W$. As in [53] final output is mean-pooled to compress the output of the last SSM layer along the sequence length dimension to enable softmax classification of the given sequence. Each sequence layer is preceded with batch-normalization. While in [53] two different learning rates for the SSM parameters and the other parameters are used, we use a single learning rate for all parameters. As in [53], we do not apply weight-decay to the SSM parameters (except for the feedthrough matrix $D$ as it is not affected by our HSV regularization). In Table 2 we show the parameters used to generate our results. The parameters for dropout, weight-decay and regularization magnitude (the scalar by which we multiply our regularizer (11)) are found via grid-search. Note that the regularization magnitude is small because it includes a sum of all the HSVs in all the different layers. Since we implement our regularizer by adding it to the softmax cross-entropy loss we use during training, this must be scaled, appropriately.

With this setup, in our `flax/nnx` implementation, it takes around two hours to train an sMNIST model, around four hours for training an IMDB model, and around six hours to train an sCIFAR model on a single H100 GPU. This is slightly higher than the train times reported in [26]; which we attribute to our slow train data input pipeline. Parameters are initialized from standard normal distributions unless stated otherwise. The SSM parameters $\rho$ are initialized using Gaussian distributions with mean 1.5, and standard deviation 0.25, which yields to an eigenvalue distribution similar to that of HiPPO matrices after discretization. Note that, as stated in Section 2.1, $\rho$ is subsequently thresholded using a $\tanh$ nonlinearity. The matrices $B$ and $C$ in each state space layer are initialized with zero-mean Gaussian distributions, with standard deviation $1/\sqrt{n^2 + m^2}$ and $1/\sqrt{n^2 + p^2}$, respectively.

## C   Postprocessing details

### C.1 Bisection algorithm for selecting ranks

We present our algorithm for selecting the ranks in the different layers in Algorithm 1. Given a total budget $r_t$, it distributes the state dimensions across the $\ell = 1, \ldots, L$ layers such that the same amount of energy is preserved in each layer. It terminates once a prescribed tolerance or maximum number of iterations is reached. In our experiments, we set the tolerance to $\epsilon = 10^{-8}$ and the maximum number of iterations to $n_{\max} = 100$.

Table 2: Hyperparameter setup in our experiments. Depth denotes the number of sequence layers, LR the learning rate, WD the weight decay, and reg. mag. the scalar by which we multiply our regularizer (11).

|        | depth | $n$ | $p = m$ | dropout | LR | batch dim. | epochs | WD | reg. mag. |
|--------|-------|-----|---------|---------|------|-----------|--------|------|-----------|
| sCIFAR | 6 | 384 | 512 | 0.2 | 0.001 | 50 | 200 | 0.3 | 0.00002 |
| sMNIST | 4 | 128 | 128 | 0.1 | 0.001 | 50 | 250 | 0.1 | 0.00001 |
| IMDB | 6 | 192 | 256 | 0.1 | 0.001 | 50 | 35 | 0.1 | 0.001 |
| PATH | 6 | 256 | 192 | 0.1 | 0.001 | 64 | 100 | 0.03 | 0.00000001 |
| PATH-X | 6 | 256 | 128 | 0.0 | 0.0001 | 16 | 30 | 0.0 | 0.00000001 |

---

**Algorithm 1** Bisection method for reduced state dimension determination

---

**Require:** sorted HSVs of the different SSMs $[\mathbf{\Sigma}_1, \mathbf{\Sigma}_2, \ldots, \mathbf{\Sigma}_\ell]$, target order $r$, tolerance $\epsilon$, maximum number of iterations $n_{\max}$

**Ensure:** truncation order for each layer

1: Normalize each HSV vector such that $\sum_{i=1}^{n} \mathbf{\Sigma}_{j,i} = 1$ for all $\mathbf{\Sigma}_1, \mathbf{\Sigma}_2, \ldots, \mathbf{\Sigma}_\ell$
2: Set $\gamma_{\min} = 0$, $\gamma_{\max} = 1$
3: Set $\gamma = (\gamma_{\min} + \gamma_{\max})/2$
4: Set $k = 0$
5: Compute $\hat{r}$ as mean of $[\mathsf{argmin}(\mathbf{\Sigma}_1 > \gamma), \mathsf{argmin}(\mathbf{\Sigma}_2 > \gamma), \ldots, \mathsf{argmin}(\mathbf{\Sigma}_\ell > \gamma)]$
6: **while** $|\hat{r} - r| > \epsilon$ and $k < n_{\max}$ **do**
7:     Set $\gamma = (\gamma_{\min} + \gamma_{\max})/2$
8:     Set $k = k + 1$
9:     **if** $\hat{r} > r$ **then**
10:         Set $\gamma_{\max} = \gamma$
11:     **else**
12:         Set $\gamma_{\min} = \gamma$
13:     **end if**
14: **end while**
15: Return reduced orders $[\mathsf{argmin}(\mathbf{\Sigma}_1 > \gamma), \mathsf{argmin}(\mathbf{\Sigma}_2 > \gamma), \ldots, \mathsf{argmin}(\mathbf{\Sigma}_\ell > \gamma)]$

---

# D  Extra Results

Table 4 evaluates the computational cost of adding our regularizer to the SSM training procedure. When adding our regularizer with our block-wise Lyapunov-solver, the training-time only increases slightly, while the naive Lyapunov-solver leads to prohibitively high training-times. We report the increase in train time relative to unregularized training. In Figure 4, we compare our approach to all methods proposed in [26].

In Table 3, we demonstrate the runtime speed up that is obtained during inference at different truncation ratios.

In Table 5, we report median and standard deviation of the test accuracies across three different training runs, in which the models are initialized with different random seeds. Note that, importantly, for truncation ratios, where the accuracy of the original model is retained (until around 80% for sCIFAR (grayscale) and sMNIST), the standard deviation is low, and it only increases after that threshold. This is because after losing approximation accuracy of the original SSM layers, the sequence-to-sequence maps change in different ways across the different runs, which has a different impact on the test accuracy.

We also compare our HSVR approach to a simple $\ell_1$-norm regularization of the diagonal blocks to justify the computational overhead incurred when computing the Hankel singular values during

Table 3: Inference runtime ratios (sCIFAR) at different truncation ratios

| trunc. ratio | 50% | 60% | 70% | 80% | 90% |
|--------------|------|------|------|------|------|
| runtime ratio | 0.66 | 0.57 | 0.51 | 0.45 | 0.40 |

Table 4: Relative runtimes measured over one training epoch. We report "−" when training is impossible due to excessive resource consumption. Remember that the costs of naive Lyapunov solver scale as $\mathcal{O}(n^6)$ Training time is measured in and extra run for one epoch to ensure the same hardware is used. Experiments are carried out on a single H100 GPU.

| regularizer | sCIFAR (grayscale) | sMNIST | IMDB |
|---|---|---|---|
| none | $1\times$ | $1\times$ | $1\times$ |
| HSVR (blocked) | $1.59\times$ | $1.12\times$ | $1.15\times$ |
| HSVR (naive) | − | $27.3\times$ | − |

Table 5: Median and standard deviation of test accuracies [%] for HSVR.

| quantity | sCIFAR (grayscale) | | | | sMNIST | | | | IMDB | | | |
|---|---|---|---|---|---|---|---|---|---|---|---|---|
| trunc. ratio | 60% | 70% | 80% | 90% | 60% | 70% | 80% | 90% | 60% | 70% | 80% | 90% |
| median | 81.53 | 81.74 | 80.28 | 27.72 | 99.28 | 99.26 | 98.95 | 60.58 | 87.32 | 87.47 | 87.40 | 85.62 |
| std. dev. | 0.20 | 0.43 | 1.22 | 2.95 | 0.02 | 0.06 | 0.22 | 16.75 | 0.13 | 0.13 | 0.15 | 0.87 |

optimization. We compare our result to a fine parameter sweep for the $\ell_1$-norm regularization magnitude on the CIFAR, MNIST, and IMDB datasets in Tables 6, 7, and 8, respectively, and observe that for almost all truncation ratios, our approach outperforms the simple $\ell_1$ regularization, which is in line with the system theoretical results for balanced truncation.

In Figure 4, we show a comparison of our method to all methods proposed in [26].

| trunc. ratio | $\ell_1, 10^{-6}$ | $\ell_1, 10^{-5}$ | $\ell_1, 10^{-4}$ | $\ell_1, 10^{-3}$ | $\ell_1, 10^{-2}$ | $\ell_1, 10^{-1}$ | HSVR |
|---|---|---|---|---|---|---|---|
| 50% | 40.17 | 29.18 | 63.02 | 65.88 | 78.86 | 72.53 | **82.19** |
| 60% | 38.05 | 28.09 | 57.53 | 61.34 | 77.15 | 69.57 | **81.84** |
| 70% | 35.35 | 24.94 | 13.56 | 55.79 | 74.84 | 64.95 | **81.75** |
| 80% | 12.28 | 21.92 | 09.95 | 45.64 | 69.90 | 52.67 | **81.37** |
| 90% | 10.76 | 09.71 | 10.76 | 30.80 | **55.54** | 31.84 | 51.08 |

Table 6: CIFAR: Test accuracies (%) in comparison to L1 regularization

| trunc. ratio | $\ell_1, 10^{-6}$ | $\ell_1, 10^{-5}$ | $\ell_1, 10^{-4}$ | $\ell_1, 10^{-3}$ | $\ell_1, 10^{-2}$ | $\ell_1, 10^{-1}$ | HSVR |
|---|---|---|---|---|---|---|---|
| 50% | 50.68 | 54.75 | 56.17 | 52.27 | 50.02 | 59.16 | **87.25** |
| 60% | 51.39 | 55.15 | 50.93 | 50.94 | 50.02 | 56.61 | **87.26** |
| 70% | 52.67 | 51.04 | 50.02 | 50.78 | 50.57 | 61.97 | **87.16** |
| 80% | 51.74 | 50.99 | 50.06 | 51.66 | 50.16 | 58.72 | **86.97** |
| 90% | 51.23 | 50.14 | 50.14 | 49.60 | 50.22 | 51.94 | **86.40** |

Table 7: IMDB: Test accuracies (%) in comparison to $\ell_1$ regularization

| trunc. ratio | $\ell_1, 10^{-6}$ | $\ell_1, 10^{-5}$ | $\ell_1, 10^{-4}$ | $\ell_1, 10^{-3}$ | $\ell_1, 10^{-2}$ | $\ell_1, 10^{-1}$ | HSVR |
|---|---|---|---|---|---|---|---|
| 50% | 58.58 | 66.94 | 98.99 | 99.01 | 99.00 | 95.88 | **99.29** |
| 60% | 39.57 | 46.28 | 97.85 | 98.54 | 98.28 | 84.92 | **99.45** |
| 70% | 13.94 | 18.88 | 86.50 | 97.53 | 97.09 | 76.81 | **99.22** |
| 80% | 12.03 | 11.03 | 14.78 | 82.56 | 90.12 | 55.09 | **98.90** |
| 90% | 11.12 | 10.62 | 9.51 | 13.61 | 37.03 | 11.72 | **86.95** |

Table 8: MNIST: Test accuracies (%) in comparison to $\ell_1$ regularization

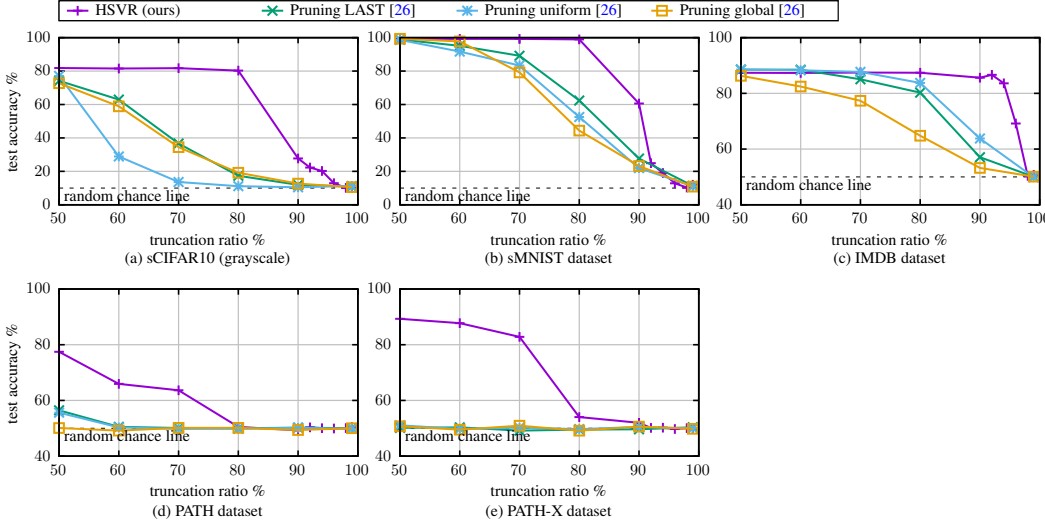

Figure 4: Comparison of HSVR to all methods in [26]

