# OpenReview forum: "Hankel Singular Value Regularization for Highly Compressible State Space Models"
_NeurIPS.cc/2025/Conference — NeurIPS 2025 poster_

### Official Review · Reviewer_6j3S · 2025-06-01

**Clarity:** 4
**Significance:** 3
**Originality:** 4
**Rating:** 5
**Confidence:** 3

**Summary:**

This is a strong paper overall, clearly written, dense with results (both mathematical and empirical).

The major contribution is to develop a new regularization approach that improves the compressibility of deep state space models (SSMs). The method achieves a 10x improvement in compressibility (compared to a baseline with no regularization).  These SSMs have become increasingly popular recently, so this work has good potential applied impact.

**Questions:**

One question re: novelty: there is a relevant paper by Forgione et al 2024 ("Model order reduction of deep structured state-space models: A system-theoretic approach") - if this isn't by the same authors, there should be some discussion of this paper.

In the abstract, the authors claim "HSVR state space layers are up to 10× more compressible than standard state space layers trained without regularization" - a more informative statement here would be to compare against the SOTA compression approach, rather than an approach with no regularization.

**Ethical Concerns:**

["NO or VERY MINOR ethics concerns only"]

**Final Justification:**

I'll stick with my original score.  I remain enthusiastic about this paper.

**Limitations:**

adequately addressed.

**Quality:**

4

**Strengths And Weaknesses:**

Repeating myself from above -

This is a strong paper overall, clearly written, dense with results (both mathematical and empirical).

The proposed method achieves a 10x improvement in compressibility.  These SSMs have become increasingly popular recently, so this work has good potential applied impact.

No major weaknesses noted.

---

> ### Author Rebuttal · Authors · 2025-07-30
>
> > There is a relevant paper by Forgione et al 2024 ("Model order reduction of deep structured state-space models: A system-theoretic approach") - if this isn't by the same authors, there should be some discussion of this paper.
> - We thank the reviewer for bringing this interesting work to our attention. Indeed the authors also follow a system-theoretic approach and regularize Hankel singular values, which is a natural idea when balanced truncation is used for compression. The main methodological difference to our work is that they use a complex-valued diagonal parameterization, whereas we build on real block-diagonal parametrization. Our block-diagonal parametrization has advantages in terms of stability (see Section 2.1) but requires a more careful approach for computing Hankel singular values efficiently (Section 2.2). We also believe that our approach achieves better compression rates because we propose an energy-bisection approach (Section 2.4) for determining the reduced rank in each layer considering the entire DNN while Forgione et al. choose the same reduced ranks in each layer. In any case, we completely agree with the reviewer that this paper is very relevant and shares the idea of regularizing the Hankel singular values. We will certainly cite this paper, update the Introduction and contributions of our paper, and provide a detailed discussion in our paper, if it gets accepted.
>
> > In the abstract, the authors claim "HSVR state space layers are up to 10x more compressible than standard state space layers trained without regularization" - a more informative statement here would be to compare against the SOTA compression approach, rather than an approach with no regularization.
> - We will extend the abstract and state that we maintain high accuracy of 99% even at high compression rates such as 80% where other compression methods have a loss of accuracy to 60% (LAST).

---

### Official Review · Reviewer_1JxP · 2025-06-29

**Clarity:** 4
**Significance:** 1
**Originality:** 3
**Rating:** 3
**Confidence:** 3

**Summary:**

The paper introduces Hankel Singular Value Regularization to improve the compressibility of state-space models.
The proposed regularization encourages rapid decay of Hankel singular values, making the SSM layers more compressible.
The authors also develop an efficient algorithm to compute this regularization during training.
Experimental evaluations demonstrate significant improvements in compressibility compared to unregularized models.

**Questions:**

How would applying your proposed truncation ratio affect performance specifically in architectures with relatively small SSM components, such as LRUs (Orvieto et al., 2023) or the Mamba architecture (Gu et al., 2023)?

**Ethical Concerns:**

["NO or VERY MINOR ethics concerns only"]

**Final Justification:**

Final justification for score:

As noted in the rebuttal thread, I remain unconvinced by the practical usability of the method. The paper targets compression of SSM components, but in realistic architectures like Mamba, these components represent a small fraction of total parameters—e.g., <4%.

Furthermore, their regularization approach assumes time-invariant dynamics, which does not hold for Mamba's selective (time-varying) SSMs.

The response did not resolve these concerns. I see potential in the direction but believe stronger evidence is needed for real-world impact.

**Limitations:**

Yes

**Paper Formatting Concerns:**

None.

**Quality:**

3

**Strengths And Weaknesses:**

**Strengths**
* Clearly written and well structured.
* Comprehensive experiments that validate the effectiveness of the proposed method.
* Offers a simple and intuitive solution.

**Weaknesses**
* The motivation for this research appears weak. Specifically, the portion of SSM parameters in contemporary architectures like Mamba (Gu et al., 2023) and other non-selective SSM variants such as LRUs (Orvieto et al., 2023) is minimal relative to the total network parameters. For instance, Mamba explicitly states most parameters reside in linear projections rather than the SSM component, which is relatively small (with hidden dimensions typically set at 16). Thus, the actual practical gain from compressing only the SSM parts may be limited.
* The paper would greatly benefit from demonstrating significant model-size reductions in realistic, widely-used architectures.

---

> ### Author Rebuttal · Authors · 2025-07-30
>
> > [...] the portion of SSM parameters in contemporary architectures like Mamba (Gu et al., 2023) and other non-selective SSM variants such as LRUs (Orvieto et al., 2023) is minimal relative to the total network parameters. [...] Thus, the actual practical gain from compressing only the SSM parts may be limited.
> - The reviewer brings up a good point: there are other parameters in network architectures that our approach does not compress. We agree with this statement. However, we also would like to point out that for example Mamba-2 (Dao, Gu, 2024) uses a state-space dimension of 256, which is similar to the sizes we use. Furthermore, also the paper that the reviewer cites (Orvieto et al., 2023; Table 10) considers models with state-space dimension in the 100s. Given these numbers from the literature and that we also use state-space dimensions of similar size, we do believe that compressing the SSM parameters can be beneficial also for realistic models.
> - In any case, if this work gets accepted, we will add a clear statement to the limitation section that our approach compresses the SSM parameters but that many models have other parameters that we do not compress.
> > The paper would greatly benefit from demonstrating significant model-size reductions in realistic, widely-used architectures.
> - We consider standard benchmark problems for long-range tasks and show compression ratios of 10x while maintaining more than 90% of the accuracy. While these are not architectures with as many parameters as, e.g., Mamba-2, we see similar dimensions of state-space models in other architectures (see previous comments) and thus believe our results indicate that our approach is promising.

---

> > ### Comment · Reviewer_1JxP · 2025-08-02
> >
> > I thank the authors for their continued engagement, but I believe they are misinterpreting my criticism and, therefore, are not addressing it adequately.
> >
> > The authors state:
> >
> > > "Mamba-2 (Dao, Gu, 2024) uses a state-space dimension of 256, which is similar to the sizes we use. Furthermore, also the paper that the reviewer cites (Orvieto et al., 2023; Table 10) considers models with state-space dimension in the 100s."
> >
> > While I agree that the dimensions of the state-space models (SSMs) used in this paper align with theoretical frameworks presented, this is not my primary concern. Specifically, my criticism arises from the fact that, as I previously mentioned in my review, Mamba explicitly states "most parameters reside in linear projections rather than the SSM component." To the best of my knowledge, SSMs are typically lightweight components (in terms of parameters) within larger neural network architectures. Consequently, compressing these small components alone might not lead to significant improvements in either memory or computation speed.
> >
> > As I previously highlighted in my review, "The paper would greatly benefit from demonstrating significant model-size reductions in realistic, widely-used architectures."
> >
> > To address this criticism adequately, the authors need to demonstrate substantial compression rates concerning the entire architecture, not just the SSM components. Without clear evidence of this broader impact, it remains impossible for me to recommend acceptance.
> >
> > __I will hold off submitting my final score for a few days to give the authors an opportunity to respond thoroughly.__

---

> ### Author Response · Authors · 2025-08-04
>
> We thank the reviewer for pointing us to the text passage that raised their concern. In the following, we provide details that even for Mamba-2 as used for Multi-Query Associative Recall, a 10x reduction in the SSM parameters as we achieve in our experiments leads to a ~71% reduction in total number of parameters (non-SSM + SSM parameters). Given these findings, the reviewer is right that there are other parameters that we do not reduce but we still believe the prospect of a 71% reduction is valuable. And that it is worth exploring methods like ours for reducing the number of SSM parameters.
>
> Here are the details. We have carefully read that paper again and compared with the official Mamba implementation on GitHub:
>
> - The authors state (Gu, Dao 2024 (Mamba); Section 3.4): “For each block, most of the parameters (3*E*D^2) are in the linear projections (2*E*D^2 for input projections, E*D^2 for output projection) while the inner SSM contributes less.” Here D is the ‘model dimension’ and ‘E’ is an expansion factor.
> - We did not find a concrete statement of the precise number of the SSM parameters per block in the main text so we checked the official implementation (GitHub: state-spaces/mamba/mamba_ssm/modules/mamba_simple.py). Therein, we indeed find the 2ED^2 parameters for the input projection (self.in_proj in line 62) and the ED^2 parameters for the output projection (self.out_proj in line 117). However, the number of SSM parameters is significant compared to that; we have (N+1)*E*D parameters for A (self.A_log (lines 104-110)) as well as (ED (dt_rank + 2N)) parameters for B,C, and Delta (collected in self.x_proj, lines 77-79). The variable dt_rank has a default value of ceil(ED/16) (rounding up ED/16 to the nearest integer). In this way we have E*D*(3*N+1+dt_rank) SSM parameters, which is not negligible, especially for larger state dimensions.
> - As an example, for E=2 (as in Mamba) D=64, N=16 (a small state dimension, as in Mamba-2 using larger state dimensions is proposed), we have 24576 non SSM parameters and 7296 SSM parameters, which is definitely smaller but not negligible. Moreover, as the state dimension N is increased (as is proposed in Mamba-2 and what is also used in most of the S5 benchmarks (with N in the 100s)) the number of SSM parameters is increased as O(N).
> - Our results indicate that a 10x compression ratio is possible without significant losses in accuracy. For E=2, D=64, N=256 (as used in Mamba-2 for Multi-Query Associative Recall), this would lead to a reduction from 24576 (non SSM) + 99,456 (SSM) to 24576 (non SSM) + 11136 (SSM) parameters (with reduced state dimension of N=26), which is a significant reduction of more than 70% in number of parameters.

---

> > ### Comment · Reviewer_1JxP · 2025-08-07
> >
> > Sorry for the late response, I missed the earlier message. I hope it's still possible to clear up any misunderstanding I might have.
> >
> > Thanks for the response. However, I still think the example used to support the 71% compression figure is misleading.
> >
> > First, the regularization method in the paper is designed for time-invariant SSMs, where the Hankel matrix is well-defined. But Mamba is a selective (i.e., time-varying) model.
> >
> > Second, the configuration used in the rebuttal (D=64, N=256) is only used in very small tasks. (The amount of parameters in this setting is very small, \~0.5M or \~3M if we use setting more like the Mamba paper) in more realistic tasks, a larger model is used. For example, in the language modeling tasks explored in the Mamba paper (Appendix E.2), the numbers are very different: D = 1024 and N = 16 (for example).This gives:
> >
> > * Non-SSM parameters: 3ED² = 3×2×1024² = 6,291,456
> > * SSM parameters = 236,544
> > * Total = 6,528,000
> >
> > Even with a 10× compression on the SSM part (down to \~23,654), total parameters only drop to \~6.31M — <**4% reduction overall**;
> >
> > That said, perhaps there are realistic settings where the method is impactful, or maybe even 3% savings is meaningful enough in practice. I think this paper has potential, but stronger evidence of real-world relevance would help.
> >
> > I’m keeping my score as is and hope the authors continue developing this direction so its value can be better appreciated.

---

> > > ### Author Response · Authors · 2025-08-08
> > >
> > > Thank you for your response.
> > > The reviewer is correct in that we designed the regularization method for time-invariant SSMs, as used (Smith et al. 2025). It is also correct that larger-scale architectures like Mamba are selective, which can be interpreted as time-varying and it may thus seem like they are not amenable to our regularization and benefit from the improved compressibility. However, it is important to note that this ‘time-invariance’ is in fact an input, output, and time-step modulation (This also becomes clear in the reference implementation with ‘mamba_simple.py’ lines 135-141 and 189-204, where input and output projections are applied before and after the inner SSM is called), for which Hankel Singular Values remain a meaningful quantity.
> > >
> > > Given the reviewer’s parameter-count example, let us stress again that in Mamba (Gu, Dao, 2024) only small state space dimensions of up to N=16 are studied, which seems to be an outlier in the Deep SSM literature. As we stated before: Mamba-2 (Dao, Gu, 2024) uses a state-space dimensions of up to N=256, and find that they outperform Mamba with N=16 significantly (cf. Figure 8 in Dao, Gu, 2024). Furthermore, also in the other paper that the reviewer cited earlier (Orvieto et al., 2023; Table 10 (or Table 9 in the arXiv version)), models with state-space dimension in the 100s are considered. As our calculation shows, for such larger state dimensions, our model compression is effective.

---

### Official Review · Reviewer_exo8 · 2025-07-01

**Clarity:** 2
**Significance:** 3
**Originality:** 3
**Rating:** 4
**Confidence:** 4

**Summary:**

This paper proposes Hankel singular value regularization (HSVR) and a post-training optimization for state space models (SSMs) parameterized with block rotation matrices, inspired primarily by the balanced truncation in the model order reduction literature. HSVR induces trained systems to have fast-decaying Hankel singular values (HSVs), meaning that the energy flow is concentrated in a small number of states, by regularizing the sum of HSVs. After training, the method performs balanced truncation, eliminating low-energy states. The block rotation matrix parameterization is introduced to enable efficient computation of HSVs during training. The method is validated on selected tasks from the Long Range Arena (LRA) benchmark.

**Questions:**

1. **(Block rotation parameterization)** Why is the rotational parameterization necessary for HSVR? In other words, why didn’t the authors simply parameterize the SSMs as diagonal continuous-time SSMs with discretization? While the rotational parameterization does have the benefit of directly ensuring a stable discrete-time SSM, isn’t it also possible that diagonal SSMs can compute HSVs efficiently in $O(n^2)$? It is thus unclear whether the claimed advantage of efficient HSV computation justifies the use of the rotational SSMs. An ablation study or a comparative analysis of the computational cost between diagonal and rotational SSMs would help clarify this point.
2. **(Balanced truncation)** If the system is not yet balanced, isn’t it appropriate to define and compute the HSVs as $\sqrt{\lambda_i(PQ)}$, not as $\sigma_i(PQ)$? However, the in-training regularization process before the balanced transformation is described using $\sigma_i(PQ)$ (e.g., Line 122, Line 182). Was this the actual implementation of HSV computation during training? Moreover, do the compressed SSMs have different parameters from the uncompressed SSMs, applied by balanced transformation?
3. **(Regularization)** In the absence of additional bias, how can one ensure that regularizing nuclear norm (penalizing the sum of HSVs) promotes fast decay in HSVs rather than uniformly reducing all HSVs? The claim that ”Nuclear norm regularization, i.e., penalizing the sum of singular values of a matrix, to encourage singular values to decay rapidly, is common practice in machine learning.” (Lines 230-231) would benefit from citation of relevant references in the machine learning literature.
4. **(Experimental verification)** Can the proposed method be applied to longer sequence modeling tasks? Specifically, while results are shown on sMNIST and selected LRA tasks (sCIFAR10, IMDB), can the result be extended to the full LRA benchmark or other long sequence datasets?
5. **(Correctness and consistency of equations)** Many **Lyapunov, Sylvester, and vectorized equations** throughout the paper appear to be written wrong or deviate from standard formulations in terms of sign (e.g., Equations (2)-(5), Lines 174-176). Moreover, the notation for $A(\text{angle}, \text{radius})$ or $A(\text{radius}, \text{angle})$ should be unified across Sections 2.1 and 2.2 for consistency.

**Ethical Concerns:**

["NO or VERY MINOR ethics concerns only"]

**Final Justification:**

My initial Weak Accept score was based primarily on the contribution and empirical effectiveness of HSVR. Although the authors have made a careful rebuttal and offered more details on the new system parameterization and experiments, I remain unconvinced that the proposed parameterization is necessary. Moreover, the experimental additions are partial. Thus, I maintain my Weak Accept.

**Limitations:**

Limitations are already addressed in the paper ((a) inapplicability to pretrained models, (b) linear compression, and (c) time-invariant models).

**Paper Formatting Concerns:**

No concern

**Quality:**

2

**Strengths And Weaknesses:**

- **Strength 1**: HSVR is grounded in the well-established balanced truncation framework, empirically demonstrating effective compression of trained SSMs.
- **Strength 2**: The block rotation matrix parameterization introduces a novel class of SSMs, which can be efficiently implemented via associative scan by leveraging the mathematical properties of rotation matrices.
- **Weakness 1**: While diagonal SSMs are the dominant choice in recent literature, the necessity of using block rotation matrix parametrization in conjunction with HSVR remains unclear. A comparison or ablation study on different parameterization methods would strengthen the claim.
- **Weakness 2**: The scope of experimental validation is limited. Only a subset of the LRA dataset was used, and there is a lack of experiments on tasks with longer sequences.
- **Weakness 3**:  Several equations throughout the paper (e.g., Lyapunov, Sylvester, and vectorized forms) raise concerns about the correctness and clarity of mathematical presentation.

---

> ### Author Rebuttal · Authors · 2025-07-30
>
> > While diagonal SSMs are the dominant choice in recent literature, the necessity of using block rotation matrix parametrization in conjunction with HSVR remains unclear. A comparison or ablation study on different parameterization methods would strengthen the claim.
> - The purpose of the block structure is to stay in real arithmetic. What we show is that in this parametrization, we can efficiently compute Hankel singular values as well. We also stress that the block rotation matrix parametrization is used for training purposes only. But once trained and compressed, we transform the system back into a diagonal system.
> > Only a subset of the LRA dataset was used, and there is a lack of experiments on tasks with longer sequences. [...] Can the proposed method be applied to longer sequence modeling tasks?
> - We demonstrate our approach on a range of examples and achieve in all experiments significant compression, outperforming other approaches. We agree that even more experiments could make our argument stronger. We therefore also computed the Path test case (see table below) We attain for example 77.46% accuracy for 50% compression ratio which outperforms the best result in [Gwak, 2024] (LAST), which is at 56.45% for the same compression ratio. Notice that in this example all methods achieve lower accuracy. Our method can shine in situations when accuracy is high, in which case we show that we can compress more than other methods (see Table 1 in the paper).
>
> | trunc. ratio | LAST | Global H-inf | Uniform H-inf | HSVR (ours) |
> | --- | --- | --- | --- | --- |
> | 50.00 | 56.45 | 50.10 | 55.61 | 77.46 |
> | 60.00 | 50.51 | 49.16 | 50.32 | 65.93 |
> | 70.00 | 50.11 | 50.15 | 49.78 | 63.64 |
> | 80.00 | 49.96 | 50.16 | 49.84 | 50.50 |
> | 90.00 | 50.25 | 49.37 | 50.16 | 49.28 |
>
> - All experiments that we conducted indicate that we achieve favorable compression ratios while maintaining accuracy. Thus, while there are more experiments we could conduct, we believe the current ones and the new additional one provide a strong indication that our approach is meaningful - additionally to the strong theoretical foundation provided by system theory.
>
> > If the system is not yet balanced, isn’t it appropriate to define and compute the HSVs as \sqrt{\lambda_i(PQ)) instead of \sigma_i(PQ). [...]
> - Yes, the reviewer is correct, sqrt(lambda(PQ))=sigma(PQ) only when PQ is normal, which is not in general the case. We are computing sqrt(lambda(PQ)) in our code and we will adapt the lines 122, 182, accordingly. We will update the text, if this work gets accepted.
>
> > [...] how can one ensure that regularizing nuclear norm (penalizing the sum of HSVs) promotes fast decay in HSVs rather than uniformly reducing all HSVs?
> - The nuclear norm is the best convex approximation of the rank function (Fazel, PhD thesis, 2002). The idea is similar to using the \ell_1 norm instead of the \ell_0 norm in sparse recovery. The nuclear norm is used, for example, for matrix completion in the famous paper by Candes and Recht (FoCM, 2009), which is essentially about solving the “netflix problem” in machine learning.
>
> > Many Lyapunov, Sylvester, and vectorized equations throughout the paper appear to be written wrong or deviate from standard formulations in terms of sign (e.g., Equations (2)-(5), Lines 174-176).  [...] notation for A(angle, radius) and A(radius, angle) should be unified across Sections 2.1 and 2.2 for consistency.
> - Yes, thank you very much for catching this. We will update the Lyapunov equations to APA^T - P + BB^T = 0 and  A^T Q A - Q + C^T C=0. We will unify the notation to A(radius, angle) for consistency.

---

> > ### Comment · Reviewer_exo8 · 2025-08-06
> >
> > I appreciate the authors’ efforts to address the concerns raised. I have a few follow-up remarks for further clarification.
> >
> > 1. **(Block rotational parameterization)** In the manuscript (e.g., Lines 132–133), the authors claim that the $O(n^2)$ HSV computation is enabled by the block rotational parameterization. However, I believe that a similar efficiency can also be achieved with a diagonal parameterization using complex conjugate pairs, rather than real-valued rotations. I would appreciate further clarification on the necessity of real-valued block parameterization in terms of computational efficiency.
> > 2. **(Balanced truncation)** The authors have addressed my concern appropriately on this point.
> > 3. **(Regularization)** I am still unclear how nuclear norm regularization leads to *selective shrinkage* of HSVs, rather than *uniform shrinkage*. If the current formulation is indeed encouraging rapid decay of HSVs, I would appreciate a more concrete explanation or hypothesis supporting this behavior.
> > 4. **(Experimental verification)** Could the authors clarify whether the additional experiment was conducted on Pathfinder or Path-X? I still believe that demonstrating performance across all LRA tasks would strengthen the claim of **broad applicability and stability** of the proposed parameterization and method, and I would recommend including such results.
> > 5. **(Correctness and consistency of equations)** The authors have adequately addressed my concern regarding the equations.

---

> > > ### Author Response · Authors · 2025-08-06
> > >
> > > 1. You are correct, a similar efficiency can be achieved with a diagonal parametrization when switching to complex arithmetic. We will add this to our paper. The statement is referring to the fact that we exploit this structure to obtain O(n^2) computation of the HSV which is much lower than the standard algorithm (for dense systems). We will add a section to the Appendix that explains these computations also for the complex diagonal case to enable the use of HSV regularization for both parameterization choices.
> > > 2. Thank you and we are happy we could address your concern.
> > > 3. The current nuclear norm regularization indeed leads to a selective shrinkage of HSVs. The intuition is as follows: While the penalization is on the sum and thus affects all HSVs uniformly, the effect is not uniform on the singular values. This is because the larger HSVs contribute to the dynamics and are thus often crucial for the overall data-fit/prediction accuracy, the smaller HSVs are not and are thus more readily shrunk. There is a large body of literature supporting this sum-of-singular-values-based regularization for encouraging sparsity or low-rank solutions. In the excellent SIAM-Review article (Recht, et al.: Guaranteed Minimum-Rank Solutions of Linear Matrix Equations via Nuclear Norm Minimization (2010)), the authors point to several references where this technique is used. Also (Mazumder et al.: Spectral Regularization Algorithms for Learning Large Incomplete Matrices (2010)) use that “under many situations the nuclear norm is an effective convex relaxation to the rank constraint”. We will add these and additional references that justify our regularizer for obtaining a rapid decay of HSVs. In particular, (Fazel, PhD thesis, 2002), Candes and Recht (FoCM, 2009), Candès and Tao (IEEE TAC, 2010), Recht et al. 2010 (SIAM Review), (Mazumder et al. JMLR 2010) and clearly mention that we build on that to use the same kind of regularization for the HSV.
> > > 4. The new results are obtained for the pathfinder dataset. We will aim to include additional results in the final version, if accepted, but in the current limited time we have, we could provide new results for the pathfinder dataset only. We agree that this is a limitation. At the same time, we point out that all results, including the new pathfinder result, are in agreement that our approach leads to a significant compression.
> > > 5. Thank you again for catching these.

---

> > > > ### Comment · Reviewer_exo8 · 2025-08-07
> > > >
> > > > The authors have carefully addressed my comments, and I appreciate their thoughtful response.
> > > >
> > > > - If the proposed block rotational parameterization is not essential for computational efficiency (compared to the currently general choice of diagonal SSMs), its advantages, such as simplicity with real-valued parameters, should be more clearly emphasized. Moreover, a direct comparison between **diagonal parameterization + HSVR** and **block rotational parameterization + HSVR** could improve the completeness of the empirical section.
> > > > - Thank you for clarifying the role of nuclear norm regularization.
> > > > - I respect the authors' acknowledgment of the limited diversity in experiments. While the proposed method consistently outperforms baselines across the included settings, expanding the experimental coverage would strengthen the persuasiveness of the work.

---

> > > > > ### Author Response · Authors · 2025-08-07
> > > > >
> > > > > 1. Thank you for this advice, we will certainly emphasize the other advantages of the block rotational parametrization in our paper. Moreover, we will look into a numerical comparison between the two approaches
> > > > > 2. We are glad we could clarify the role of the nuclear norm regularizer.
> > > > > 3. That is a valid point. We are actively working on the other datasets right now. We have run an experiment on the challenging Path-X dataset as well (we had to reduce the learning rate by a factor of 10 to make it work and only trained for half the epochs reported in (Smith et al. 2023) to get the results in time for the rebuttal period but these indeed look promising, as we can again drastically improve the compressibility of the SSM compared to the unregularized approach and compared to previous methods for SSM compression. Please find our preliminary new results on PATH-X, below. Thank you again for encouraging us to run these additional experiments. If the paper gets accepted, we will report these results (and a comparison to the diagonal case) to the experimental section.
> > > > >
> > > > > PATH-X
> > > > > | trunc. ratio | LAST | Global H-inf | Uniform H-inf | BT no regularization | HSVR (ours) |
> > > > > | -- | -- | -- | -- | -- | -- |
> > > > > | 50% | 50.16 | 50.71 | 51.08 | 58.57 | 89.30 |
> > > > > | 60% | 50.33 | 49.50 | 49.74 | 56.09 | 87.74 |
> > > > > | 70% | 49.16 | 50.93 | 50.23 | 50.39 | 82.81 |
> > > > > | 80% | 49.53 | 49.14 | 49.70 | 50.16 | 54.02 |
> > > > > | 90% | 49.64 | 50.61 | 50.47 | 50.13 | 51.97 |

---

### Official Review · Reviewer_3qrK · 2025-07-01

**Clarity:** 3
**Significance:** 2
**Originality:** 2
**Rating:** 4
**Confidence:** 3

**Summary:**

This work (HSVR) discussed Hankel singular value regularization (HSVR) for state space models (SSM) in order to make SSMs more compressible. The author first introduced a regularizer based on the Hankel singular value, which aims to make the model more compressible while balancing the task accuracy. The authors studied how to efficiently compute HSVR under the rotation matrices parameterized SSMs as previously proposed in the HiPPO framework. Showing the final output error bound is upper limited by terms involving summation of Hankel singular values across layers, the authors discussed how to compress with balanced truncation across different layers in a single model. Empirical experiments were conducted on several long-range arena tasks and showed that the HSRV method outperformed a major previous SSM regularization method and no regularization baselines at different truncation ratios.

**Questions:**

Questions:

1. The weight decay rate selected in Table 2 is quite different from the LAST paper. Were the LAST results in this work using the same weight decay as the HSVR method? How does it interact with HSVR for different weight decay?

2. Line 251, what is $\u_k$? Should it be input $x_k$?. And for completeness, could you provide proof for equation 12?

3. While applying tanh on the vector \rou could enforce stability, does it hurt the performance? Is there an ablation study supporting the benefits?

**Ethical Concerns:**

["NO or VERY MINOR ethics concerns only"]

**Final Justification:**

This work proposed a different parameterization schema as [1], which resulted in a better compression rate than [1], despite the fact that  they both inspect SSM through the Hankel operator. Considering the significance of these 2 works, I am raising the score by 1 point. The authors should include a clear comparison and citation of [1] in the final version if upon acceptance.

Thanks to the author for the additional experimental result. However, similar to Reviewer exo8’s comment, the experimental evaluation is rather limited compared to works alike.

**Limitations:**

Yes: the authors have adequately addressed the limitations and potential negative societal impact of their work

**Paper Formatting Concerns:**

No concerns

**Quality:**

3

**Strengths And Weaknesses:**

Strengths:
1. Deducted the formulation for computing the Hankel singular values, exploiting the block-diagonal structure, which brings down the complexity from $n^3$ to $n^2$.

2. The work discussed the compatibility and complexity of the proposed methods in the SSM paradigm, including the regularizer, associative scan, and post-training compression.

3. Heuristic results on limited tasks show a promising truncation-performance trade-off, superior to one major baseline.

Weaknesses:

1. The author did not compare to the work [1], which studies the SSM initialization and training stability with the Hankel operator. The authors also developed a parameterization scheme that utilizes Markov parameters within Hankel operators. Considering the close relationship between this work and [1], a detailed comparison in theory and experiments should be included. If the authors could articulate the difference and provide an experimental comparison, I would consider raising the score for originality and overall score.

2.. LAST covered more experiments from the long range arena. It would be nice to also include them in the benchmarks, especially for Pathfinder and Path-X. Otherwise, there are in total 3 tasks from LRA that have been empirically examined, which is quite lacking. From the training hours provided in the LAST paper, the training resources needed for Pathfinder and PATH-X are on the same level as sCIFAR.

3. The method is a model pruning/ compression method. But the author did not show real inference speed up or theoretical FLOPs numbers.


[1] HOPE for a robust parameterization of long-memory state space models, Yu et al. 2025 ICLR https://openreview.net/pdf?id=RZwtbg3qYD    on arxiv in 2024 https://arxiv.org/pdf/2405.13975

---

> ### Author Rebuttal · Authors · 2025-07-30
>
> > The author did not compare to the work (Yu et al. 2025 ICLR), which studies the SSM initialization and training stability with the Hankel operator. [...]  Considering the close relationship between this work and (Yu et al. 2025 ICLR), a detailed comparison in theory and experiments should be included. [...]
> - We thank the reviewer for sharing this interesting work with us. The work cited by the reviewer introduces a new parametrization of SSMs which achieves compression of 3x of the parameters (from $3n^2$ to $n^2$) of a regular S4D/S4 model. However, we stress that we achieve compression rates of 10x (1/10 of the parameters of the uncompressed model) with our approach. We achieve these higher rates because we not only use a favorable parametrization as (Yu et al. 2025 ICLR) but additionally introduce a regularization that nudges the optimization to find compressible models.
> - We will cite this work and add this comparison to our paper, if our work gets accepted.
>
> > It would be nice to also include [...] Pathfinder and Path-X.
> - We are adding results for the Path example to the paper. We attain for example 77.46% accuracy for 50% compression ratio which outperforms the best result in [Gwak, 2024] (LAST), which is at 56.45% for the same compression ratio. Notice that in this example all methods achieve lower accuracy. Our method can shine in situations when accuracy is high, in which case we show that we can compress more than other methods (see Table 1 in the paper).
>
> | trunc. ratio | LAST | Global H-inf | Uniform H-inf | HSVR (ours) |
> | --- | --- | --- | --- | --- |
> | 50% | 56.45 | 50.10 | 55.61 | 77.46 |
> | 60% | 50.51 | 49.16 | 50.32 | 65.93 |
> | 70% | 50.11 | 50.15 | 49.78 | 63.64 |
> | 80% | 49.96 | 50.16 | 49.84 | 50.50 |
> | 90% | 50.25 | 49.37 | 50.16 | 49.28 |
>
> > [...] author did not show real inference speed up or theoretical FLOPs numbers.
> - We conducted new experiments and we can report runtime improvements in the appended table, which shows the runtime ratio alongside the retained accuracy for the sCIFAR example. For example, for the 80% truncation, we maintain an accuracy of 81.37 but reduce the inference runtime by 55%. We will add these new results as well as for the other examples on inference runtime speedups in the paper, if it gets accepted.
>
> | trunc. ratio | HSVR (accuracy) | HSVR (runtime ratio) |
> | --- | --- | --- |
> | 50.00 | 82.19 | 0.66 |
> | 60.00 | 81.84 | 0.57 |
> | 70.00 | 81.75 | 0.51 |
> | 80.00 | 81.37 | 0.45 |
> | 90.00 | 51.08 | 0.40 |
>
> > Were the LAST results in this work using the same weight decay as the HSVR method? How does it interact with HSVR for different weight decay?
> - We have extracted the LAST results directly from the LAST paper so the setup is exactly the same and was not modified. In our experiments, using weight-decay prevented the model from overfitting.
>
> > Line 251, what is u_k? Should it be input x_k?
> - We denote with u_k the input and with x_k the state. The seq-to-seq map goes from input u_k to output y_k, where x_k is the internal state. It is correctly written on line 251.
>
> > And for completeness, could you provide proof for equation 12?
> - Proving the error bound of balanced truncation is involved, but we will include a reference to Section 7.2.1 in A.C. Antoulas, Approximation of Large-Scale Dynamical Systems, SIAM, 200, which shows a proof.
>
> > While applying tanh on the vector \rou could enforce stability, does it hurt the performance? Is there an ablation study supporting the benefits?
> - Training without using tanh here is very unstable because the optimizer easily lands in regions where the \rho becomes numerically infinite and the optimization cannot recover. Thus, an ablation study shows very poor performance without tanh.

---

### Official Review · Reviewer_1ZYF · 2025-07-02

**Clarity:** 3
**Significance:** 2
**Originality:** 3
**Rating:** 3
**Confidence:** 4

**Summary:**

This paper proposes Hankel Singular Value Regularization (HSVR) for compressing SSM layers. The regularization leads to a decay in the singular values of the Hankel matrix and effectively reduces the dimensionality of the SSM layer. The authors develop an algorithm to compute Hankel singular values during training using block-diagonal rotation matrices, and demonstrate compression improvements on a subset of tasks from the Long Range Arena benchmark.

**Questions:**

- Have the authors considered how their method compares to a simple sparsity regularization over diagonal SSMs?
- Can the authors please address the issue of the practical tradeoff of using the method suggested? Specifically discussing the overhead of the Hankel computation and the performance speedup.

**Ethical Concerns:**

["NO or VERY MINOR ethics concerns only"]

**Final Justification:**

The authors have satisfied my requests and provided further experiments supporting their claims.

**Limitations:**

Please see points and concerns raised above.

**Paper Formatting Concerns:**

No concerns regarding formatting issues.

**Quality:**

2

**Strengths And Weaknesses:**

### Strengths:

- Important Problem: The paper addresses the relevant issue of sequential model compression, which is crucial as these models scale to larger applications.
- Novel Theoretical Approach: The connection between Hankel singular values and SSM compressibility is an interesting theoretical topic with ties to classical system theory.
- Technical Soundness: The manuscript appears technically sound with rigorous mathematical development. Specifically, the contribution of an efficient algorithm for computing Hankel singular values.
- Clear Presentation: The paper is generally well-written with clear exposition of the methodology and theoretical background.


### Major Weaknesses
1. Missing Critical Baseline Comparison - The most significant weakness is the lack of comparison to L1 regularization on diagonal SSMs. This represents a fundamental gap in the experimental evaluation. The most obvious comparison is to diagonal SSMs which have been studied and showed to achieve comparable performance to the original SSM architecture (Diagonal State Spaces are as Effective as Structured State Spaces; Gupta et al. NeurIPS 2022).

The fact that this comparison is missing raises fundamental concerns regarding the relevance of the suggested method. L1 regularization is the most natural and obvious approach for encouraging sparsity/compressibility in diagonal systems, and the paper provides no justification for why the proposed complex rotation-based parametrization and Hankel singular value computation is necessary when a simple L1 penalty on diagonal elements could achieve similar compression goals.
The authors must either:
- Demonstrate that L1 regularization on diagonal SSMs fails to achieve comparable compression
- Provide theoretical arguments for why their approach is fundamentally superior
- Acknowledge this as a major limitation

Without this comparison, the contribution appears over-engineered for the problem at hand.

2. Practical considerations and applicability:
- Computational overhead: The paper does not include any analysis of training time costs from computing Hankel singular values during optimization. Is this method practical?
- Gains from compression: What is the practical application the suggested method should lead to? The authors could attempt to tie the Hankel regularization to the problem of finding the minimal degree necessary to solve a problem to a desired accuracy. As it stands, without any additional context, it is not clear if the suggested method brings any benefit without analyzing the overhead incurred by the regularization with the speedup gained by the compression.

---

> ### Author Rebuttal · Authors · 2025-07-30
>
> > The most significant weakness is the lack of comparison to L1 regularization on diagonal SSMs.
> - Experiments: We now conducted experiments with L1 regularization; see tables with new results below. In all experiments, our approach outperforms compression with L1 regularization. For high compression of 90%, the accuracy of the model using L1 regularization is consistently 20 percentage points below what our HSVR models achieve. See tables below.
>
> CIFAR:
>
> | trunc. ratio | L1 1e-5 | L1 1e-3 | L1 1e-1 | HSVR |
> | --- | --- | --- | --- | --- |
> | 50% | 29.18 | 65.88 | 72.53 | 82.19 |
> | 60% | 28.09 | 61.34 | 69.57 | 81.84 |
> | 70% | 24.94 | 55.79 | 64.95 | 81.75 |
> | 80% | 21.92 | 45.64 | 52.67 | 81.37 |
> | 90% | 9.71 | 30.80 | 31.84 | 51.08 |
>
> Imdb:
>
> | trunc. ratio | L1 1e-5 | L1 1e-3 | L1 1e-1 | HSVR |
> | --- | --- | --- | --- | --- |
> | 50% | 54.75 | 52.27 | 59.16 | 87.25 |
> | 60% | 55.15 | 50.94 | 56.61 | 87.26 |
> | 70% | 51.04 | 50.78 | 61.97 | 87.16 |
> | 80% | 50.99 | 51.66 | 58.72 | 86.97 |
> | 90% | 50.14 | 49.60 | 51.94 | 86.40 |
>
> MNIST:
>
> | trunc. ratio | L1 1e-5 | L1 1e-3 | L1 1e-1 | HSVR |
> | --- | --- | --- | --- | --- |
> | 50% | 66.94 | 99.01 | 95.88 | 99.29 |
> | 60% | 46.28 | 98.54 | 84.92 | 99.45 |
> | 70% | 18.88 | 97.53 | 76.81 | 99.22 |
> | 80% | 11.03 | 82.56 | 55.09 | 98.90 |
> | 90% | 10.62 | 13.61 | 11.72 | 86.95 |
> - Theory: From system theory we know that fast decaying Hankel singular values are determining whether an LTI system is compressible or not (line 50 in the paper). This means that while L1 regularization can work too, our approach is backed up by the strong mathematical foundation of system theory. In particular, balanced truncation (which we use to compress our HSVR models) comes with the strong guarantee given by the error bound in (12). In contrast, regularization with L1 regularization and subsequent pruning/truncation can be related to ‘modal truncation’ in the system theory, which again is a viable approach but does not explicitly try to keep the error of the input-output (seq-to-seq) map low as eq (12).
> - We will add the new experiments to the paper, should this paper get accepted.
> > No analysis of the training time computational overhead. “Is this method practical?”
> - In Table 3 in the appendix we show that HSVR with the proposed solver incurs costs of 1.12x to 1.59x of the costs of having no regularization, which indicates that the training overhead is manageable.
> - We stress that the training overhead is so low because of the algorithms that we developed in this work (Section 2.2). Off-the-shelf techniques are orders of magnitude slower (as we also report in Table 3).
> > No demonstration of gains from compression
> - First, we note that compression alone already reduces storage requirements. This can be useful if networks are meant to be embedded in large applications.
> - Second, there is an inference runtime speedup because the evaluation becomes cheaper when there are fewer parameters. We conducted new experiments and we can report runtime improvements in the appended table, which shows the runtime ratio alongside the retained accuracy for the sCIFAR example. For example, for the 80% truncation, we maintain an accuracy of 81.37 but reduce the inference runtime by 55%. We will add these new results as well as for the other examples on inference runtime speedups in the paper, if it gets accepted.
>
> | trunc. ratio | HSVR (accuracy) | HSVR (runtime ratio) |
> | --- | --- | --- |
> | 50% | 82.19 | 0.66 |
> | 60% | 81.84 | 0.57 |
> | 70% | 81.75 | 0.51 |
> | 80% | 81.37 | 0.45 |
> | 90% | 51.08 | 0.40 |

---

> > ### Comment · Reviewer_1ZYF · 2025-08-04
> > **Request for Expanded Baseline Evaluation and Real-World Applicability Analysis**
> >
> > Thank you for the response and for providing additional experiments.
> >
> > I remain unconvinced that the comparison to the L1 baseline is entirely fair. The evaluation appears limited to only three coefficient values; I would expect a more thorough sweep over a wider range to ensure the conclusions are robust.
> >
> > Regarding applicability and impact, I find the scope of the presented compression schemes still limited. While I acknowledge that extended experiments may be beyond the rebuttal phase, I would like to see a detailed discussion of realistic deployment scenarios, including expected speedups and compression ratios for full-scale models that incorporate both FFN and SSM blocks.
> >
> > If the authors engage in a substantive discussion on these points, I am open to updating my score accordingly.

---

> > > ### Author Response · Authors · 2025-08-04
> > >
> > > Thank you again for asking about the L1 baseline. As we agree that this is an important comparison to make, we have extended the parameter sweep to the values 1e-6,1e-5,...,1e-1 for each example; see tables below. We maintain the result that ‘our Hankel Singular Value regularization leads to more compressibility than L1 regularization’ even after the finer parameter sweep. We thank the reviewer again for bringing up this baseline, it definitely strengthened our paper by comparing to it.
> > >
> > > CIFAR (fine sweep):
> > > | trunc. ratio | L1 1e-6 | L1 1e-5 | L1 1e-4 | L1 1e-3 | L1 1e-2 | L1 1e-1 | HSVR |
> > > | --- | --- | --- | --- | --- | --- | --- | --- |
> > > | 50% | 40.17 | 29.18 | 63.02 | 65.88 | 78.86 | 72.53 | 82.19 |
> > > | 60% | 38.05 | 28.09 | 57.53 | 61.34 | 77.15 | 69.57 | 81.84 |
> > > | 70% | 35.35 | 24.94 | 13.56 | 55.79 | 74.84 | 64.95 | 81.75 |
> > > | 80% | 12.28 | 21.92 | 09.95 | 45.64 | 69.90 | 52.67 | 81.37 |
> > > | 90% | 10.76 | 09.71 | 10.76 | 30.80 | 55.54 | 31.84 | 51.08 |
> > >
> > > Imdb (fine sweep):
> > > | trunc. ratio | L1 1e-6 | L1 1e-5 | L1 1e-4 | L1 1e-3 | L1 1e-2 | L1 1e-1 | HSVR |
> > > | --- | --- | --- | --- | --- | --- | --- | --- |
> > > | 50.00 | 50.68 | 54.75 | 56.17 | 52.27 | 50.02 | 59.16 | 87.25 |
> > > | 60.00 | 51.39 | 55.15 | 50.93 | 50.94 | 50.02 | 56.61 | 87.26 |
> > > | 70.00 | 52.67 | 51.04 | 50.02 | 50.78 | 50.57 | 61.97 | 87.16 |
> > > | 80.00 | 51.74 | 50.99 | 50.06 | 51.66 | 50.16 | 58.72 | 86.97 |
> > > | 90.00 | 51.23 | 50.14 | 50.14 | 49.60 | 50.22 | 51.94 | 86.40 |
> > >
> > > MNIST (fine sweep):
> > > | trunc. ratio | L1 1e-6 | L1 1e-5 | L1 1e-4 | L1 1e-3 | L1 1e-2 | L1 1e-1 | HSVR |
> > > | --- | --- | --- | --- | --- | --- | --- | --- |
> > > | 50.00 | 58.58 | 66.94 | 98.99 | 99.01 | 99.00 | 95.88 | 99.29 |
> > > | 60.00 | 39.57 | 46.28 | 97.85 | 98.54 | 98.28 | 84.92 | 99.45 |
> > > | 70.00 | 13.94 | 18.88 | 86.50 | 97.53 | 97.09 | 76.81 | 99.22 |
> > > | 80.00 | 12.03 | 11.03 | 14.78 | 82.56 | 90.12 | 55.09 | 98.90 |
> > > | 90.00 | 11.12 | 10.62 | 09.51 | 13.61 | 37.03 | 11.72 | 86.95 |
> > >
> > > With regards to the impact on larger models, we have worked out the number of SSM and feed-forward parameters for the popular MAMBA architecture (based on the official implementation at github) and concluded that our compression ratios of around 10x at which we still retain high performance, lead to a reduction of more than 70% of the total number of parameters (non-SSM and SSM parameters): The number of SSM parameters per layer is given by E*D*(3*N+1+dt_rank) in the Mamba architecture (E: expansion factor (typically 2-3), D: model dimension (typically 32-256 in Mamba-2), N: state dimension (varies between 16-256 in Mamba-2). If we consider the setup E=2, D=64, N=256 (as used in on of the examples in Mamba-2), this would lead to a reduction from 24576 (non SSM) + 99,456 (SSM) to 24576 (non SSM) + 11136 (SSM) parameters (with reduced state dimension of N=26), which is a significant reduction of more than 70% in the total number of parameters. (Details in the response to reviewer 1JxP.)
> > >
> > > We ask the reviewer to let us know if we can provide additional information about the performance of our approach.

---

> > > > ### Comment · Reviewer_1ZYF · 2025-08-09
> > > > **Score update**
> > > >
> > > > I appreciate the additional experiments and your efforts to address my concerns.
> > > > I have revised my score accordingly.

---

### Decision · Program_Chairs · 2025-09-17

**Decision:**

Accept (poster)

**Comment:**

The authors present a method for regularizing deep SSMs with time-invariant dynamics to use lower dimensional hidden states. Their method is based on well-established principles for model order reduction based on the Hankel singular values, but they show how those singular values can be computed efficiently for SSMs with block diagonal dynamics matrices. The paper addresses an interesting and important topic, the presentation is clear, and the results are compelling.

Some reviewers leaned towards rejecting the paper on the grounds that it: 1) did not originally compare against L1 regularized diagonal dynamics matrices (1ZYF); 2) does not address linear time-varying layers like Mamba (1JxP); and 3) may not be as impactful in models where SSM layers constitute a smaller fraction of total parameter count (1JxP). I think the authors did a good job of rebutting the first criticism, and while I think the 2nd and 3rd points are valid, the strengths of this paper outweigh those limitations. Indeed, three other reviewers found this paper to be a significant contribution worthy of acceptance, and I agree with them.

Minor comments:
- Lines 35-38 imply that $u_k \in \mathbb{R}^p$, but you also seem to use $m$ for the input dimension. I would suggest introducing the inputs as $m$-dimensional and then noting (as you do later) that you assume $m=p$ for analysis, since that is the common design choice.
- Reviewer exo8 made a reasonable point that the block diagonal parameterization of a real-valued A matrix looks equivalent to a diagonal complex-valued A matrix since the eigenvalues come in complex conjugate pairs. Since readers will likely be familiar with the complex diagonal parameterization in S5, I would suggest making this correspondence more clear.
-  Reviewer 6j3S mentioned an important related work of Forgione et al '24. Please cite and discuss the relationship to your work in the final manuscript.